# Enhancers regulate 3′ end processing activity to control expression of alternative 3′UTR isoforms

Buki Kwon [1], Mervin M. Fansler [1,2], Neil D. Patel[1], Jihye Lee [1], Weirui Ma[1] & Christine Mayr [1,2 ✉]

Multi-UTR genes are widely transcribed and express their alternative 3′UTR isoforms in a cell type-specific manner. As transcriptional enhancers regulate mRNA expression, we investigated if they also regulate 3′UTR isoform expression. Endogenous enhancer deletion of the multi-UTR gene *PTEN* did not impair transcript production but prevented 3′UTR isoform switching which was recapitulated by silencing of an enhancer-bound transcription factor. In reporter assays, enhancers increase transcript production when paired with single-UTR gene promoters. However, when combined with multi-UTR gene promoters, they change 3′UTR isoform expression by increasing 3′ end processing activity of polyadenylation sites. Processing activity of polyadenylation sites is affected by transcription factors, including NF-κB and MYC, transcription elongation factors, chromatin remodelers, and histone acetyltransferases. As endogenous cell type-specific enhancers are associated with genes that increase their short 3′UTRs in a cell type-specific manner, our data suggest that transcriptional enhancers integrate cellular signals to regulate cell type-and condition-specific 3′UTR isoform expression.

[1] Cancer Biology and Genetics Program, Memorial Sloan Kettering Cancer Center, New York, NY 10065, USA. [2] Tri-Institutional Training Program in Computational Biology and Medicine, Weill Cornell Graduate College, New York, NY 10021, USA. ✉email: mayrc@mskcc.org

Most of our knowledge on gene expression regulation has been gained through analysis of genes that generate mRNAs with constitutive 3′UTRs, meaning that their pre-mRNAs are processed into mRNA isoforms with single 3′UTRs. This class of genes contains widely expressed housekeeping genes but also the majority of developmentally regulated genes whose transcription is switched on or off in a cell type-specific manner[1–3]. Cell type-specific gene expression is known to be regulated by transcriptional enhancers[1,2,4–6]. However, approximately half of human genes use alternative cleavage and polyadenylation (APA) to generate mRNA isoforms that encode the same protein but differ in their 3′UTR sequence[3]. The majority of these genes are widely expressed, but they are characterized by tissue- and cell type-specific expression of specific 3′UTR isoforms. These genes are enriched in regulatory factors, including transcription factors, RNA-binding proteins, kinases, and ubiquitin enzymes[3]. However, their mode of regulation is largely unknown, and it is currently unclear how cell type-specific expression of individual mRNA isoforms with unique 3′UTRs is achieved.

APA is developmentally regulated and can be dysregulated in disease[7,8]. Inclusion of different regulatory elements in 3′UTRs influences mRNA stability, translation, and localization[9,10]. A difference in 3′UTR sequence can also determine protein function as alternative 3′UTRs allow newly translated proteins to participate in alternative protein complexes[11–15]. Alternative 3′UTR isoform usage is regulated by RNA-binding proteins, including polyadenylation and splicing factors, but also by factors that influence transcription elongation[16–25]. Knock-down of polyadenylation factors often changes 3′UTR isoform usage of hundreds of genes, but genome-wide analyses of 3′UTR isoform expression across cell types and conditions point to a more fine-grained regulation of APA[3,7,8]. It is currently largely unknown how alternative 3′UTRs of individual genes are regulated in a gene- and condition-specific manner.

According to the original definition, transcriptional enhancers are DNA sequences that increase the expression of a reporter gene[4–6,26]. Currently, increased gene expression is often used interchangeably with increased transcription, thus implying that enhancers mostly affect transcript production[6,27]. However, the generation of mature mRNAs requires pre-mRNA production and processing which includes splicing and 3′ end cleavage and polyadenylation (CPA)[8,28]. Therefore, when disregarding the contribution of mRNA stability, mRNA production of unspliced transcripts is largely determined by the number of produced transcripts and by the 3′ end processing activity that we call here CPA activity.

Under physiological conditions, it is currently difficult to disentangle the contribution of transcript production and transcript processing to the expression level of single-UTR genes. However, viral infection or osmotic stress impair the transcript processing of cellular genes at a large scale. This leads to massive read-through transcription downstream of polyadenylation signals (PAS), thus illustrating the crucial contribution of transcript processing[29–32]. Moreover, point mutations or genetic variants that occur in PAS or in their surrounding sequence elements reveal the contribution of 3′ end processing activity to mRNA expression[33–40]. Such mutations result in 1.5 to 2-fold differences in steady-state mRNA levels which is sufficient to cause disease phenotypes, including thalassemia, thrombophilia, or cancer predisposition[33–35,37,38,40].

Here, we set out to investigate if cell type- or condition-specific expression of 3′UTR isoforms is regulated by transcriptional enhancers. PTEN is a multi-UTR gene and we found that deletion of the endogenous PTEN enhancer did not reduce transcript production but impaired CPA activity at a proximal and intrinsically weak PAS. The enhancer-dependent processing activity regulation was mediated by transcription factors, transcription elongation factors, and chromatin modifiers. Enhancer-mediated regulation of 3′UTR isoform expression is widespread as endogenous, cell type-specific enhancers significantly associate with genes that exclusively upregulate their 3′UTR isoform expression in a cell type-specific manner. Our data indicate that transcriptional enhancers regulate both aspects of mature mRNA generation, namely transcript production and 3′ end processing to regulate mRNA and mRNA isoform expression in a cell type- and condition-specific manner.

## Results

**The PTEN enhancer induces a 3′UTR isoform switch of endogenous PTEN.** PTEN is a tumor-suppressor gene whose expression is altered in a large fraction of cancers. Cells are very sensitive to PTEN dosage as even a small decrease in PTEN expression is cancer-promoting[41]. The PTEN gene generates multiple mRNA isoforms with alternative 3′UTRs that encode the same protein. In our previous 3′UTR isoform expression study, PTEN was among the top genes with extensive differences in alternative 3′UTR isoform usage across cell lines and tissues[3]. To obtain a better understanding of PTEN expression regulation, we applied CRISPR technology to delete the promoter-proximal PTEN enhancer in the breast cancer cell line MCF7 which expresses wild-type PTEN[42]. The boundaries of the PTEN enhancer were determined using ChIP-seq data on transcription factor binding sites and acetylated H3K27 levels (Fig. 1a)[43–46]. We used a pair of guide RNAs to delete the PTEN enhancer. We obtained two control clones with the wild-type (WT) enhancer sequence and two "delta enhancer" (dE) clones with a heterozygous deletion in the region of the PTEN enhancer (Fig. 1b and Supplementary Fig. 1a, b).

Heterozygous deletion of the enhancer increased steady-state PTEN mRNA level by only 1.17-fold and did not affect protein level (Fig. 1c, d). We hypothesized that enhancer activation may be necessary to observe an effect. The PTEN enhancer contains canonical NF-κB binding sites (Fig. 1a)[47]. As cytoplasmic acidification was previously reported to increase NF-κB activity in MCF7 cells[48], we cultivated the cells in acidified media (pH = 6.5) and measured NF-κB activity by blotting for phosphorylated transcription factor p65 (which is encoded by the RELA gene). Although cultivation of cells in acidified media increased phosphorylated p65 (Fig. 1e and Supplementary Fig. 1c), we did not observe an enhancer-mediated difference in PTEN mRNA and protein levels (Fig. 1c, d).

Promoters were previously implicated in the regulation of mRNA processing[49–59]. Therefore, we investigated if deletion of the enhancer would change alternative 3′UTR isoform expression of PTEN. Enhancer deletion had little effect on 3′UTR isoform expression under normal cultivation conditions (Fig. 1f and Supplementary Fig. 1d). However, in acidified conditions, we observed a striking switch in 3′UTR isoform expression with increased expression of the short 3′UTR (SU) isoform of PTEN which was fully abrogated in cells with heterozygous deletion of the enhancer (Fig. 1f and Supplementary Fig. 1d).

As the PTEN enhancer overlaps with the KLLN gene, we tested the influence of the KLLN gene on PTEN 3′UTR isoform expression. shRNA-mediated knock-down (KD) of the KLLN gene did not influence PTEN mRNA or 3′UTR isoform expression (Supplementary Fig. 1e–g). This supports our result that the PTEN enhancer is responsible for the 3′UTR isoform change. As cultivation of MCF7 cells in acidified media activates NF-κB, we tested if KD of the transcription factor p65 influences PTEN 3′UTR isoform expression. KD of p65 did not affect

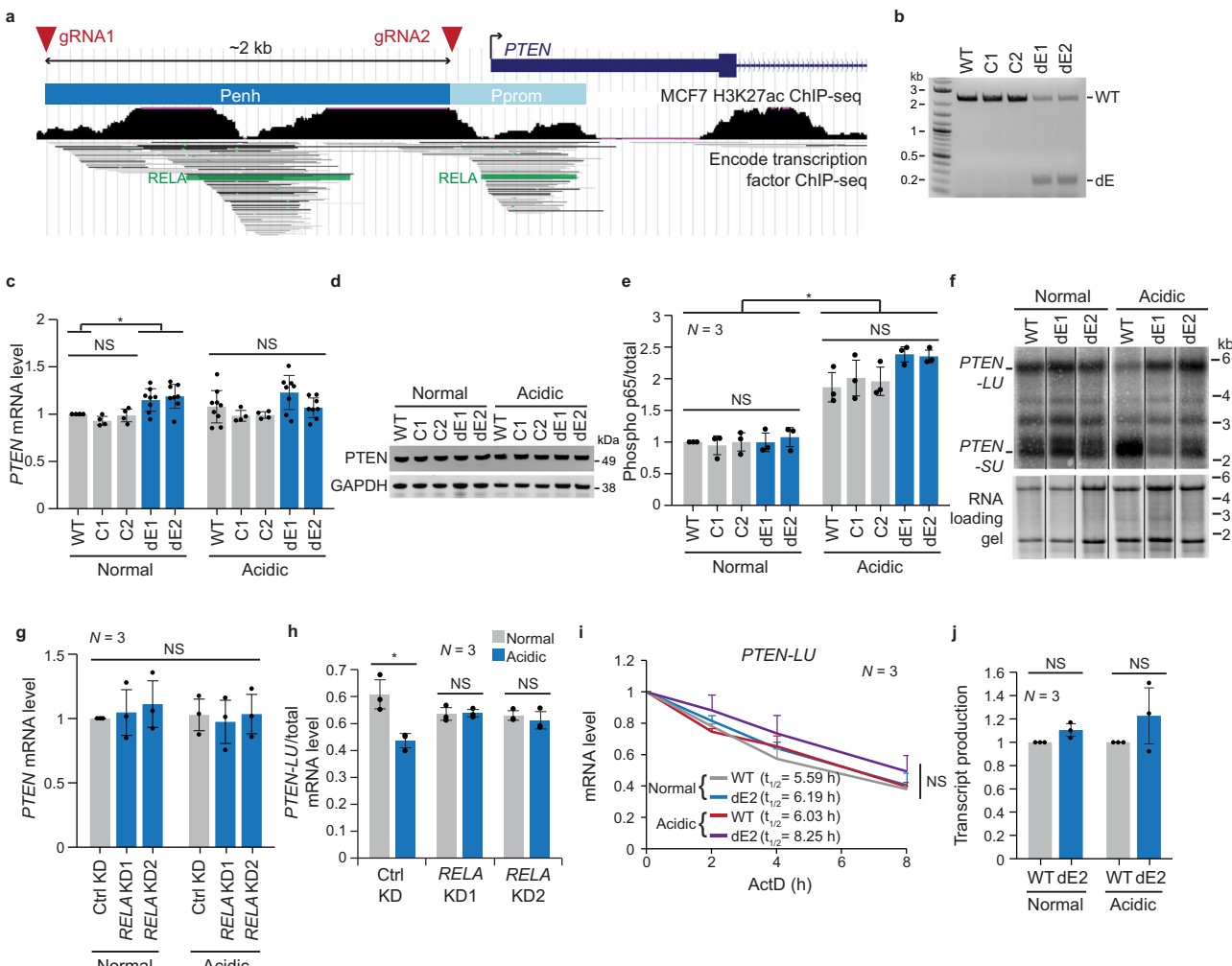

**Fig. 1 The *PTEN* enhancer induces a 3′UTR isoform switch of endogenous *PTEN*. a** UCSC genome browser snapshot showing the *PTEN* genomic locus around the transcriptional start site (arrow). The *PTEN* enhancer (Penh) was deleted using the indicated guide RNAs (red arrow heads). Among transcription factor binding sites identified by ChIP-seq, RELA binding sites are highlighted. *PTEN* promoter, Pprom. **b** Genotyping PCR was performed in parental MCF7 cells (WT), wild-type clones (C1, C2), and heterozygous enhancer deletion clones (dE1, dE2) with a primer pair flanking the deleted region. Shown is an agarose gel with the indicated PCR products. $n = 3$ biologically independent experiments. **c** *PTEN* mRNA expression measured by RT-qPCR in the indicated samples. Data are shown as mean ± std. of $n = 4$ biologically independent experiments for C1 and C2 and $n = 8$ biologically independent experiments for WT, dE1, and dE2 after normalization to *RPL19*. One-way ANOVA with Tukey's post-hoc test was performed. $*P = 0.0048$ between WT and dE1 and $P = 0.0015$ between WT and dE2; NS, not significant. **d** Representative western blot showing steady-state PTEN protein levels in the indicated samples. GAPDH serves as loading control. $n = 3$ biologically independent experiments. **e** Steady-state levels of phosphorylated p65 (S536) and total p65 were determined by western blot for the indicated samples and normalized to the levels of WT in the normal condition. The fraction of phosphorylated p65 over total p65 is shown as mean ± std of $n = 3$ biologically independent experiments. One-way ANOVA with Tukey's post-hoc test was performed. $*P = 5 \times 10^{-9}$; NS not significant. **f** Representative northern blot showing *PTEN* mRNA isoforms in the indicated samples. The RNA gel is shown as loading control. *SU* short 3′UTR; *LU* long 3′UTR. $n = 3$ biologically independent experiments. **g** As in (C), but for ctrl KD and *RELA* KD samples. Data are shown as mean ± std of $n = 3$ biologically independent experiments. One-way ANOVA was performed. NS not significant. **h** Steady-state levels of *PTEN-LU* and total *PTEN* mRNA were measured by RT-qPCR in the indicated samples. The fraction of *PTEN-LU* over total *PTEN* mRNA is shown as mean ± std of $n = 3$ biologically independent experiments. Two-tailed t-test for independent samples was performed. $*P = 0.0156$; NS not significant. **i** Quantification of *PTEN-LU* mRNA expression at four time points after inhibition of transcription with actinomycin D (ActD) in the indicated samples. The values were obtained by RT-qPCR, normalized to the 0 h time point and are shown as mean ± std of $n = 3$ biologically independent experiments. Half-life ($t_{1/2}$) of *PTEN-LU* mRNA is shown for each sample. One-way ANOVA was performed at each time point. NS not significant. **j** Metabolic labeling with 4-thiouridine (4sU) was used to enrich newly transcribed mRNAs. The newly transcribed RNAs were thiol-alkylated and biotinylated, followed by Streptavidin pull-down. The fraction of newly transcribed over total *PTEN* transcripts is shown for the indicated samples and was measured using RT-qPCR with a primer pair in the first intron. Data are shown as mean ± std of $n = 3$ biologically independent experiments. Two-tailed *t*-test for independent samples was performed. NS not significant. Source data for figures (**b–j**) are provided as a Source Data file.

steady-state *PTEN* mRNA levels, but it abrogated the switch in 3′ UTR isoform expression (Fig. 1g, h and Supplementary Fig. 1h). This result demonstrates that a transcription factor that binds to the *PTEN* enhancer is responsible for the 3′UTR isoform change of endogenous *PTEN*.

Mechanistically, the switch in 3′UTR isoform expression is either caused by a change in alternative PAS usage or by preferential degradation of the long 3′UTR (*LU*) isoform. However, degradation of the *LU* isoform would need to be accompanied by increased transcript production to account for

the upregulated *SU* expression. To identify the mechanism by which the enhancer controls the switch in 3′UTR isoform expression, we measured transcript production and stability. We inhibited transcription with actinomycin D and performed northern blot and qRT-PCR analysis at different time points to measure stability of the alternative 3′UTR isoforms. We did not detect an enhancer- or condition-specific difference in stability of the mRNA isoforms (Fig. 1i and Supplementary Fig. 1i). Next, we used metabolic labeling with 4-thiouridine to examine an enhancer-dependent change in transcript production before and after media acidification. We did not observe a significant difference in *PTEN* pre-mRNA production between WT and dE mutant cells or between normal and acidified conditions (Fig. 1j). Enhancers are further known to regulate alternative splicing[58], but we did not observe enhancer-dependent alternative splicing of *PTEN* (Supplementary Fig. 1j). As the *PTEN* enhancer was required for a switch in 3′UTR isoform expression without substantially regulating transcript production or causing differential stability of the alternative mRNA transcripts, our data suggest that it regulates CPA activity.

**The *PTEN* enhancer increases CPA activity of intrinsically weak PAS in a reporter system**. Processing activity of endogenous transcripts cannot be fully disentangled from transcript production and stability. To investigate if transcriptional enhancers indeed control CPA activity, we developed a reporter system. Our luciferase reporter system allows us to separately investigate enhancer-dependent transcript production and transcript processing. The PAS derived from the SV40 late transcript is one of the strongest known PAS[60,61]. When used for termination of a luciferase reporter construct it results in processing of all produced transcripts (Fig. 2a)[60,61]. Therefore, the SV40 PAS reporter construct measures transcriptional activity of the promoter, a system that has been widely used to measure transcriptional activity[62]. To assess enhancer-dependent CPA activity, we measured luciferase activity of a reporter construct that is nearly identical except that it is terminated by the proximal PAS (PPAS) of *PTEN* instead of the SV40 PAS. As only processed transcripts contribute to luciferase activity, the ratio of luciferase activities obtained from the PTEN PPAS reporter over the SV40 PAS reporter represents the relative CPA activity of the PTEN PPAS when driven by a specific promoter (Fig. 2a). In this reporter system, CPA activity corresponds to luciferase activity if the different polyadenylation sites do not influence stability of the reporters. To minimize the elements that may affect mRNA stability we used minimal PAS that only contain ~100 base pairs of surrounding sequence to ensure proper 3′ end processing (Supplementary Table 1a)[28].

We measured transcriptional activity of the *PTEN* promoter (Pprom) and observed a slight decrease in transcriptional activity (1.6-fold) in the presence of the enhancer (Penh-Pprom; Fig. 2b). The decrease may be due to the high transcriptional activity of the isolated *PTEN* promoter and the fact that some transcription factors act as repressors. However, typical enhancers increase transcriptional activity[6,26]. When we analyzed the effect of the *PTEN* enhancer in the context of two weaker core promoters, it indeed increased transcriptional activity, indicating that it acts as a transcriptional enhancer following the original definition (Supplementary Fig. 2a, b and Supplementary Table 1b)[6,26]. Intriguingly, the addition of the enhancer increased luciferase activity of the PTEN PPAS reporter 4-fold (Fig. 2c).

To investigate if the increase in luciferase activity is the result of an enhancer-dependent increase in CPA activity, we measured transcript abundance upstream and downstream of the PAS which enables detection of potential differences in the levels of read-through transcripts in the four reporter constructs (Fig. 2d). We compared read-through transcripts of a pair of reporters that are transcribed from the same promoter. We observed a similar amount of read-through transcripts downstream of the SV40 PAS and the PTEN PPAS when the reporters were transcribed from the Penh-Pprom constructs (Fig. 2e). However, we observed a three-fold higher amount of read-through transcripts downstream of the PTEN PPAS compared to the SV40 PAS when the reporters were transcribed from the Pprom promoter (Fig. 2e). This result supports enhancer-dependent regulation of CPA activity of the weak PTEN PPAS.

We performed additional control experiments to gain a better understanding of enhancer-dependent reporter regulation. We integrated the reporter into the genome using Flp-in cells and also observed a significant increase in CPA activity in the presence of the *PTEN* enhancer (Supplementary Fig. 2c, d). We did not observe enhancer- or PAS-dependent differences in mRNA stability of the transfected reporters, indicating that the difference in luciferase activity correlates with CPA activity (Supplementary Fig. 2e). The addition of the enhancer did not change the transcription start site of the reporter (Supplementary Fig. 2f), indicating that the mature mRNAs produced from the *PTEN* promoter in the presence or absence of the *PTEN* enhancer are identical. As circular plasmids were transfected, we measured rolling-circle transcription, as this may affect transcriptional or processing activity. We did not observe a significant difference in rolling-circle transcription of the plasmids that contain or lack the enhancer (Supplementary Fig. 2g). Enhancers regulate transcription independently of their orientation and can be located up- or downstream of genes[6,46]. When we placed the reverse complement of the *PTEN* enhancer downstream of the PAS, it enhanced transcription and PAS cleavage (Supplementary Fig. 2h–j). Taken together, these data suggest that regulation of 3′ end processing activity is a bona fide activity of transcriptional enhancers.

Next, we assessed if the enhancer controls processing activity of additional PAS (Supplementary Table 1a). We tested two PPAS (derived from *NUDT21* and *DICER1*), two distal PAS (DPAS; derived from *PTEN* and *NUDT21*), and two PAS derived from housekeeping genes (*GAPDH* and *UBC*) that generate constitutive 3′UTRs[3]. The rules that determine the intrinsic strength of PAS were unknown at the time of PAS selection[63,64]. However, the chosen proximal PAS are supposed to be weak as they contribute to less than 30% of *SU* isoform expression in most cell types analyzed[3]. PAS strength of distal or single-UTR genes cannot be inferred from isoform expression, but we expected them to be stronger as these PAS need to make sure that mature mRNAs are produced which is consistent with PAS scores estimated by a neural network[40,63,64].

We observed that the *PTEN* enhancer increased CPA activity of different proximal PAS by up to 3.6-fold (Fig. 2f). The enhancer also influenced CPA activity of two out of four non-proximal PAS (Fig. 2f). However, the activity change of non-proximal PAS was not consistent across them and cannot be predicted with our current knowledge. Therefore, we did not include them in further experiments. Our observations suggest that CPA activity of weak PAS may be low in the absence of an enhancer, but their processing activity can increase in vivo when transcribed from promoters with active enhancers.

**Promoter types determine if enhancers regulate transcription of CPA activity**. We then asked if other enhancers are also capable of regulating CPA activity and set out to test the enhancer of the *NUDT21* gene. The *NUDT21* gene encodes an important polyadenylation factor that changes CPA of hundreds of genes,

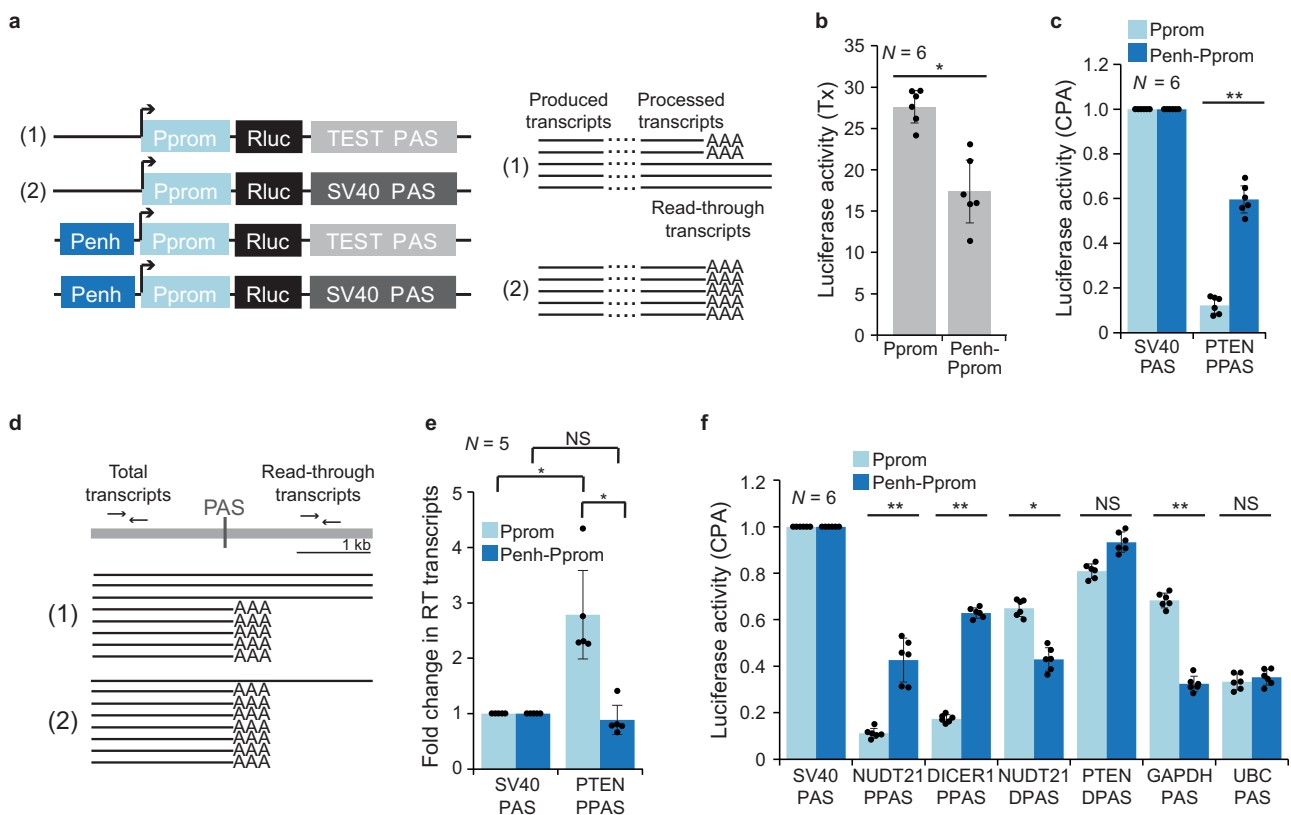

**Fig. 2 The *PTEN* enhancer increases CPA activity of proximal PAS. a** Schematic of luciferase reporter constructs to investigate enhancer-dependent transcriptional activity and CPA activity. The transcription start site is indicated by the arrow. Rluc, Renilla luciferase. Schematic showing the expected produced and processed reporter transcripts for the indicated constructs. In reporter (2), the strong SV40 PAS cleaves all produced transcripts (indicated by AAA to denote a poly(A) tail) and measures transcriptional activity. In reporter (1) a weaker PAS does not cleave all produced transcripts, thus resulting in read-through transcripts. The relative CPA activity of a test PAS corresponds to the ratio of the luciferase activities obtained from the test PAS reporter over the SV40 PAS reporter when transcribed from the same promoter. **b** Transcriptional activity (Tx) of the Pprom reporter in the presence (Penh-Pprom) or absence (Pprom) of the *PTEN* enhancer obtained by luciferase activity of the SV40 PAS reporters. Transcriptional activity represents renilla luciferase activity that was normalized by firefly luciferase activity. Shown is mean ± std of $n = 6$ biologically independent experiments. Tx, transcription. Two-tailed t-test for independent samples was performed. *$P = 0.002$. **c** Luciferase activity corresponding to the relative CPA activity of the PPAS of PTEN when transcribed from the *PTEN* promoter in the absence or presence of the *PTEN* enhancer. Shown is mean ± std of $n = 6$ biologically independent experiments. Two-tailed t-test for independent samples was performed. **$P = 1 \times 10^{-8}$; NS not significant. **d** Schematic for measuring read-through transcription of the reporter constructs shown in (**a**). A primer pair located upstream of the PAS measures the total number of transcripts produced, whereas a primer pair located downstream of the PAS measures the number of read-through transcripts. **e** Fold change in read-through (RT) transcripts obtained from the indicated reporter constructs. Shown is mean ± std of $n = 5$ biologically independent experiments. Two-tailed *t*-test for independent samples was performed. Pprom: SV40 vs PTEN PPAS *$P = 0.002$; Penh-Pprom: SV40 vs PTEN PPAS, $P = 0.42$; PTEN PPAS: Pprom vs Penh-Pprom, *$P = 0.002$. **f** As in (**c**), but additional PAS are shown. DPAS, distal PAS. Shown is mean ± std of $n = 6$ biologically independent experiments. Two-tailed *t*-test for independent samples was performed. **$P = 1 \times 10^{-6}$; *$P = 0.001$; NS not significant. Source data for figures (**b**, **c**, **e**, and **f**) are provided as a Source Data file.

when knocked-down[18,20]. Moreover, the *NUDT21* gene undergoes APA and similar to *PTEN*, its alternative 3′UTR isoforms are extensively regulated across samples[3]. We searched for ChIP-seq peaks with high H3K27 acetylation level in the vicinity of the *NUDT21* gene, as high H3K27 acetylation levels are characteristic for enhancers[6,46]. H3K27 acetylation levels in the promoter-proximal region of *NUDT21* were only intermediate, but we detected a region with very high acetylation levels 80 kb downstream of the *NUDT21* gene. We cloned 2 kb of this region and called it distal enhancer (Denh; Fig. 3a, b and Supplementary Table 1c) as we currently have no evidence that this region is an enhancer of the *NUDT21* gene. To test if the distal enhancer is functional, we measured enhancer-dependent transcriptional activity of the *GAPDH* promoter (Gprom) which drives expression of a single-UTR gene[3]. The distal enhancer upregulated transcriptional activity by more than 4-fold, thus acting as a transcriptional enhancer (Fig. 3c). However, the distal enhancer had no influence on CPA activity in the context of the Gprom

(Fig. 3d). Similar results were obtained when using the *PTEN* enhancer in the context of the Gprom (Supplementary Fig. 2b, k).

In contrast, in the context of two promoters derived from multi-UTR genes, such as the *PTEN* or *NUDT21* promoters (Nprom), the distal enhancer did not significantly affect transcriptional activity (Fig. 3b, e), but instead increased CPA activity of the PTEN PPAS between 3.4 and 5.3-fold (Fig. 3f, g) and CPA activity of the NUDT21 PPAS by 1.8-fold (Supplementary Fig. 2l, m). As the distal enhancer did not affect processing activity of a stronger PAS, our results indicate that transcriptional enhancers regulate CPA activity of weak proximal polyadenylation sites in the context of multi-UTR promoters (Fig. 3f, g).

**Transcription factors and transcription elongation factors are widespread regulators of CPA activity.** As KD of p65 changed 3′ UTR isoform expression of the endogenous *PTEN* gene, we set out to identify additional transcription factors that may mediate

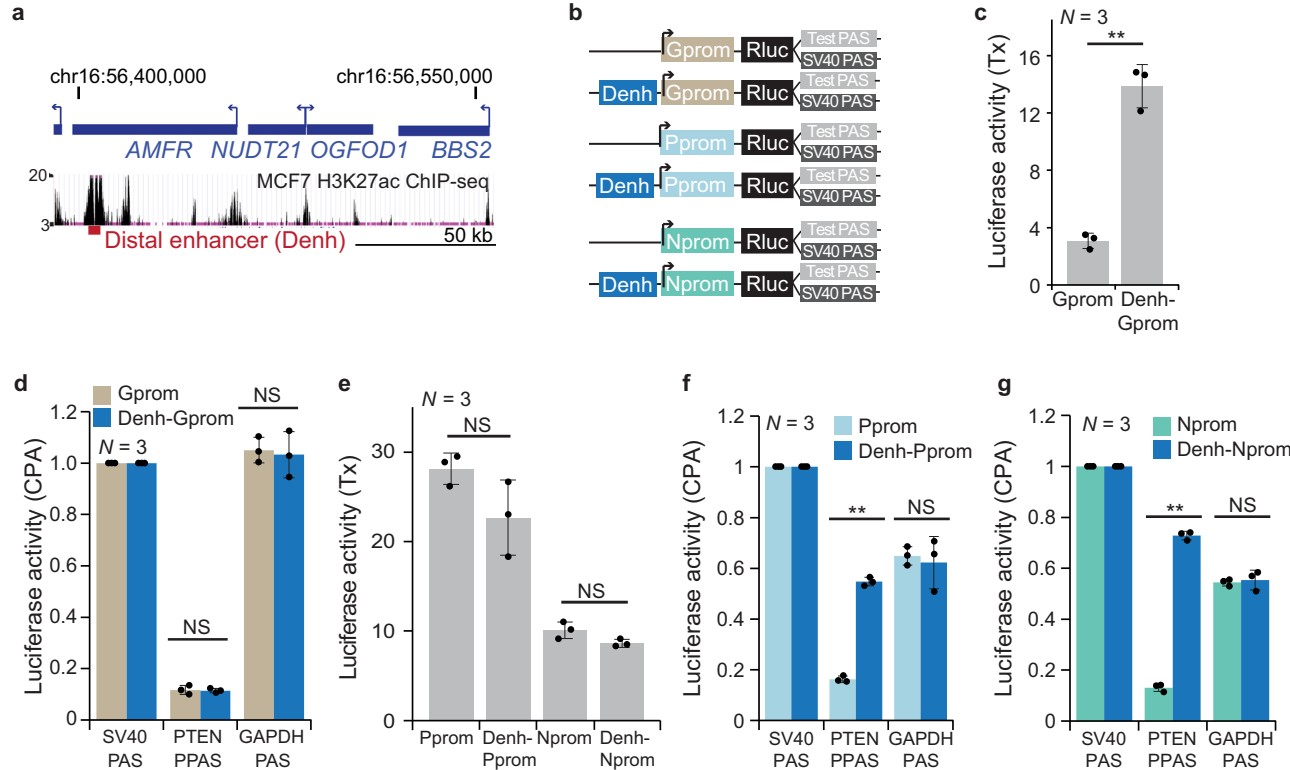

**Fig. 3 A distal enhancer regulates CPA activity of proximal PAS. a** UCSC genome browser snapshot showing the genomic context of the *NUDT21* gene locus. The region with the local maximum of acetylated H3K27 measured by ChIP-seq in MCF7 cells was defined as distal enhancer (Denh). **b** Schematic of reporter constructs used to investigate enhancer-dependent CPA activity in the context of three promoters. The *GAPDH* promoter (Gprom) drives a single-UTR gene, whereas the Pprom and *NUDT21* (Nprom) promoters drive multi-UTR genes. Shown as in Fig. 2a. **c** Enhancer-dependent transcriptional activity of the Gprom shown as in Fig. 2b. Data are shown as mean ± std of $n = 3$ biologically independent experiments. Two-tailed *t*-test for independent samples was performed. **$P = 0.0005$. **d** Enhancer-dependent CPA activity in the context of the Gprom shown as in Fig. 2c. Data are shown as mean ± std of $n = 3$ biologically independent experiments. Two-tailed *t*-test for independent samples was performed. NS, not significant. **e** As in (**c**), but enhancer-dependent transcriptional activity of two multi-UTR gene promoters is shown. Data are shown as mean ± std of $n = 3$ biologically independent experiments. Two-tailed *t*-test for independent samples was performed. NS not significant. **f** As in (**d**), but enhancer-dependent CPA activity in the context of the Pprom is shown. Data are shown as mean ± std of $n = 3$ biologically independent experiments. Two-tailed *t*-test for independent samples was performed. **$P = 1 \times 10^{-8}$. **g** As in (**d**), but enhancer-dependent CPA activity in the context of the Nprom is shown. Data are shown as mean ± std of $n = 3$ biologically independent experiments. Two-tailed *t*-test for independent samples was performed. **$P = 1 \times 10^{-5}$. Source data for figures (**c–g**) are provided as a Source Data file.

enhancer-dependent regulation of CPA activity. Binding of transcription factors to enhancers results in the recruitment of co-activators to promoters which includes components of the mediator complex, the general transcription machinery, chromatin remodelers, transcription elongation factors, and histone acetyltransferases[65–67]. We performed a small-scale shRNA screen and knocked-down individual transcription factors or co-activators expressed in MCF7 cells (Table 1 and Supplementary Fig. 3). All sequence-specific transcription factors that were knocked-down were shown by ChIP-seq to bind to the *PTEN* enhancer (Fig. 4a)[44]. We measured transcriptional and CPA activity in the context of the Penh-Pprom in control (ctrl) KD and transcription factor KD samples (Fig. 4b). As positive control, we knocked-down the CPA factor FIP1L1, which was shown previously to be required for PPAS usage[17]. KD of FIP1L1 decreased PTEN PPAS usage from 0.6 to 0.36 without affecting transcriptional activity (Fig. 4c, Table 1, and Supplementary Fig. 4a, b).

Seventeen out of 22 tested sequence-specific transcription factors significantly changed CPA activity (Fig. 4c and Table 1). Four of them appear to be CPA repressors as their KD increased CPA activity, but suppression of the majority of transcription factors decreased CPA activity of the PTEN PPAS (Fig. 4c).

Several transcription factors predominantly changed CPA activity rather than transcriptional activity in this assay. They include RELA (NF-κB p65), MYC, RXRA, and FOX1 (Fig. 4c and Table 1). KD of general transcription factors led to a binary pattern as their suppression only affected transcriptional activity or strongly repressed CPA activity, as was observed for TFIIF, TBP, TAF1, and MED1 (Fig. 4c and Table 1). Downregulation of histone acetyltransferases, chromatin remodelers, or transcription elongation factors mostly repressed CPA activity and the repressive effect was similar in strength to KD of the CPA factor FIP1L1 (Fig. 4c and Table 1). Taken together, we found that a large fraction of tested transcription factors, transcription elongation factors, and chromatin modifiers significantly influences CPA activity in the context of a reporter.

**Mutation of MYC-binding sites in the enhancer decreases CPA activity.** RELA and MYC KD also decreased CPA activity of additional PPAS but did not significantly affect cleavage activity of the strong PTEN DPAS (Fig. 4d and Supplementary Fig. 4c). Similar results were obtained after mutation of the two highly conserved MYC-binding sites in the *PTEN* enhancer (Fig. 4d, e). Mutation of the MYC-binding sites had no significant influence on transcriptional activity (Supplementary Fig. 4d), did not affect

**Table 1 Transcription factors and co-activators that regulate CPA activity of the proximal PAS of PTEN.**

| Knock-down of factor | Factor class | CPA activity | Fold repression Tx activity | Knock-down of factor | Factor class | CPA activity | Fold repression Tx activity |
|---|---|---|---|---|---|---|---|
| Ctrl1 | CTRL | 0.60 | 1.00 | EP300 | HAT | 0.32 | 3.43 |
| Ctrl2 | CTRL | 0.60 | 1.09 | KAT5 | HAT | 0.36 | 3.40 |
| FIP1L1 | CTRL | 0.35 | 0.83 | EP400 | HAT | 0.38 | 1.95 |
| RFX5 | TF | 0.20 | 2.24 | ACTL6A | HAT | 0.40 | 2.67 |
| EGR1 | TF | 0.21 | 2.05 | KAT2B | HAT | 0.41 | 1.89 |
| IRF1 | TF | 0.23 | 2.47 | ING3 | HAT | 0.45 | 2.28 |
| NFYB | TF | 0.24 | 2.79 | MORF4L | HAT | 0.46 | 1.84 |
| RELA | TF | 0.24 | 1.62 | BRD8 | HAT | 0.47 | 1.94 |
| TFAP2A | TF | 0.24 | 2.64 | RUVBL1 | HAT | 0.52 | 2.25 |
| ELF1 | TF | 0.25 | 2.61 | KAT2A | HAT | 0.62 | 3.51 |
| RXRA | TF | 0.26 | 1.38 | SIN3A | DAC | 0.88 | 2.25 |
| FOXA1 | TF | 0.30 | 1.71 | NPM1 | CHR | 0.23 | 3.24 |
| YY1 | TF | 0.36 | 2.20 | SPT16 | CHR | 0.33 | 2.59 |
| TCF12 | TF | 0.37 | 2.62 | SMARCC | CHR | 0.44 | 2.21 |
| JUND | TF | 0.37 | 1.83 | SSRP1 | CHR | 0.45 | 1.77 |
| MYC | TF | 0.44 | 1.56 | CTCF | CHR | 0.59 | 3.58 |
| JUNB | TF | 0.52 | 2.61 | RAD21 | CHR | 0.60 | 2.11 |
| CEBPB | TF | 0.55 | 3.48 | SMC3 | CHR | 0.76 | 3.50 |
| REST | TF | 0.56 | 2.34 | BARD1 | ELF | 0.27 | 1.80 |
| TFAP2C | TF | 0.57 | 3.29 | NELFE | ELF | 0.29 | 3.35 |
| E2F1 | TF | 0.69 | 2.34 | BRCA1 | ELF | 0.31 | 1.39 |
| CDKN1A | TF | 0.75 | 3.24 | SPT4 | ELF | 0.35 | 2.28 |
| JUN | TF | 0.75 | 1.62 | SPT5 | ELF | 0.36 | 3.31 |
| FOS | TF | 0.77 | 2.67 | CDC73 | ELF | 0.45 | 1.38 |
| GABPA | TF | 0.86 | 3.38 | NELFB | ELF | 0.46 | 1.73 |
| TBP | GTF | 0.19 | 2.55 | TCERG1 | ELF | 0.46 | 2.17 |
| TAF1 | GTF | 0.21 | 1.54 | USP22 | ELF | 0.46 | 2.34 |
| MED1 | GTF | 0.22 | 2.10 | CDK9 | ELF | 0.46 | 1.67 |
| GTF2F1 | GTF | 0.23 | 1.16 | | | | |
| TAF7 | GTF | 0.55 | 2.77 | | | | |
| MED12 | GTF | 0.65 | 2.76 | | | | |
| TAF12 | GTF | 0.69 | 2.50 | | | | |

*TF* Transcription factor, *GTF* General transcription factor, *HAT* Histone acetyltransferase, *DAC* Deacetylase, *CHR* Chromatin remodeler, *ELF* Transcription elongation factor.

CPA activity of the stronger distal PAS of PTEN, but decreased CPA activity of weaker proximal PAS, thus phenocopying the effect of MYC KD (Fig. 4d). These results suggest that binding of transcription factors to conserved motifs in the *PTEN* enhancer regulates CPA activity of weak proximal PAS.

**CPA activity is regulated by active enhancers.** Enhancer activation often results in histone acetylation and KD of histone acetyltransferases such as TIP60 and PCAF (encoded by *KAT5* and *KAT2B*, respectively) interferes with enhancer activation[65]. Their shRNA-mediated suppression reduced CPA activity of several proximal PAS in the context of the Penh-Pprom reporter (Supplementary Fig. 4e). KD of histone acetyltransferases also decreased CPA activity in the context of the distal enhancer but had no effect on CPA activity in reporters that lack the enhancers (Supplementary Fig. 4f–i). These results suggest that regulation of CPA activity of proximal PAS by active enhancers has the potential to be widespread as two out of two tested enhancers regulated 3′ end processing activity in the context of reporters.

**Cell type-specific enhancers preferentially associate with genes that upregulate *SU* isoforms in a cell type-specific manner.** Next, we set out to investigate if endogenous enhancers are widespread regulators of CPA activity. The analysis of multi-UTR genes allows us to distinguish transcriptional and CPA activity. Transcriptional upregulation will increase *SU* and *LU* isoform expression to a similar extent, whereas increased CPA activity of proximal PAS will only increase *SU*, but not *LU* isoform expression.

To examine if cell type-specific enhancers are associated with changes in 3′UTR isoform expression, we used a dataset that mapped erythroblast-specific enhancers and associated them with individual genes[68]. To identify genes that become upregulated in erythroblasts, we compared gene expression between erythroblasts and hematopoietic stem cells[69,70]. We analyzed single- and multi-UTR genes separately, but as expected, we observed that genes that increase their expression in erythroblasts preferentially associate with enhancers that are active in erythroblasts (Fig. 5a, b)[71]. The association with erythroblast-specific enhancers was not observed for genes whose expression did not increase (Fig. 5a, b).

Next, we focused on all multi-UTR genes and separated them into two groups based on their gene expression change between erythroblasts and hematopoietic stem cells. The two groups ("gene up", "gene not up") were further subdivided to identify among them the genes with significantly increased *SU* isoform expression ("SU up" with upregulated *SU* isoform counts and upregulated *SU* isoform ratio). For the four groups, we visualized their gene expression changes, their *SU* and *LU* isoform expression changes, and their 3′UTR isoform ratio changes (Fig. 5c–e). Next, we associated the four groups with cell type-specific enhancers that are active in erythroblasts (Fig. 5f). As expected, the groups with increased gene expression were significantly associated with cell type-specific enhancers (Fig. 5f). Importantly, also genes with exclusive upregulation of their *SU* isoforms without leading to a significant upregulation in overall gene expression were significantly associated with cell type-specific enhancers (Fig. 5f).

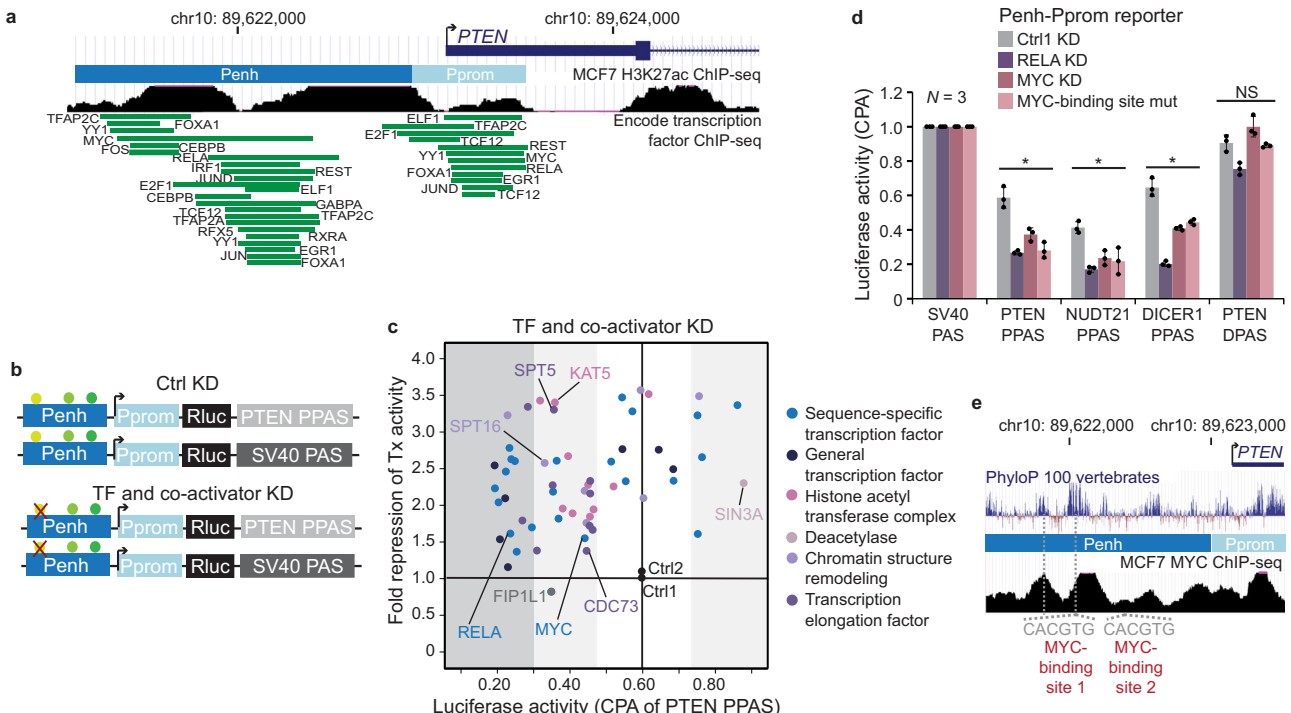

**Fig. 4 Transcription and transcription elongation factors are widespread regulators of CPA activity. a** As in Fig. 1a but shown are the binding sites of transcription factors that were knocked-down individually. **b** Schematic of reporter constructs to identify transcription factors (TFs) and co-activators that regulate CPA activity in the context of the Penh-Pprom reporter. Ctrl KD, control knock-down. **c** Summary of PTEN PPAS CPA activity and fold repression of Tx activity obtained in the shRNA screen. Shown is the mean in ctrl KD and transcription factor KD samples for the Penh-Pprom reporter. The values are reported in Table 1 and the replicates are shown in Supplementary Fig. 4a, b. The shaded areas denote the extent of change in CPA activity in TF KD samples (white, change is not significant; gray, significant change; dark gray, more than 0.30 change in CPA activity). **d** CPA activity of additional PAS when transcribed from the Penh-Pprom reporter. Shown is mean ± std of $n = 3$ biologically independent experiments after transcription factor KD or mutation of MYC-binding sites (MYC-binding site mut) in the Penh. Two-tailed $t$-test for independent samples was performed; *$P < 0.03$; NS not significant. Source data are provided as a Source Data file. **e** UCSC genome browser snapshot showing the PTEN gene locus with ChIP-seq data for MYC and the sequence conservation track of 100 vertebrates. The position of the two conserved MYC-binding sites (canonical E-boxes) are indicated.

To fully control for the slightly increased gene expression in the "SU up" group compared to the "gene not up" group, we performed stratified random sampling. To make sure that the genes in the control group do not upregulate their *SU* isoforms, we excluded all genes with a slight upregulation of *SU* from the control group, thus strongly reducing the number of genes available for comparison (Supplementary Fig. 5a–d). For three stratified random samples, our analysis shows that the fraction of genes associated with erythroblast-specific enhancers is lower in the control group (mean, 12.2%) than in the "SU up" group (mean, 15.4%), but the Chi-square test does not reach statistical significance (Supplementary Fig. 5d). This analysis revealed that it is difficult to disentangle increased gene expression from exclusive *SU* upregulation as this usually also slightly upregulates gene expression.

**Model of enhancer-mediated control of 3′UTR isoform expression**. It is well-known that active enhancers upregulate gene expression of single-UTR genes in a cell type-specific manner (Fig. 5g, top)[71]. In contrast, multi-UTR genes are usually transcribed in the majority of cell types[3]. We found that enhancer activation of multi-UTR genes can have three potential outcomes. It can result in upregulation of gene expression with *SU* and *LU* isoforms increasing similarly. It can result in a combination of change in gene and isoform expression or it can result in upregulation of *SU* isoforms without a change in gene expression (Fig. 5g, bottom).

## Discussion

Mature mRNA production is determined by the extent of transcript production and transcript processing. Here, we show that transcriptional enhancers regulate both stages of mature mRNA production, but they differentially control them for different classes of genes. At single-UTR genes, enhancers increase transcript production, whereas at multi-UTR genes, transcriptional enhancers increase transcript production or transcript processing, thus resulting in a gene expression change, in a cell type- or condition-specific 3′UTR isoform expression change or in a change of both parameters.

With our newly developed reporter assay, we were able to measure separately the two parameters of mRNA expression. Our results are consistent with the expression pattern of endogenous single- and multi-UTR genes. Whereas single-UTR genes are often transcribed in a cell type-specific manner, meaning that their expression is "off" in some cell types and "on" in others, most multi-UTR genes are always "on" as they are transcribed in the majority of cell types[3,72]. Despite their near ubiquitous expression, they encode regulatory factors and show cell type-specific 3′UTR isoform expression[3]. This expression pattern was mirrored in the reporter assays, where two transcriptional enhancers increase transcript production when the reporter gene is driven by a single-UTR gene promoter. In contrast, the same enhancers did not affect overall transcription but instead increased the expression of a transcript terminated by a weak polyadenylation site when the reporter gene was driven by

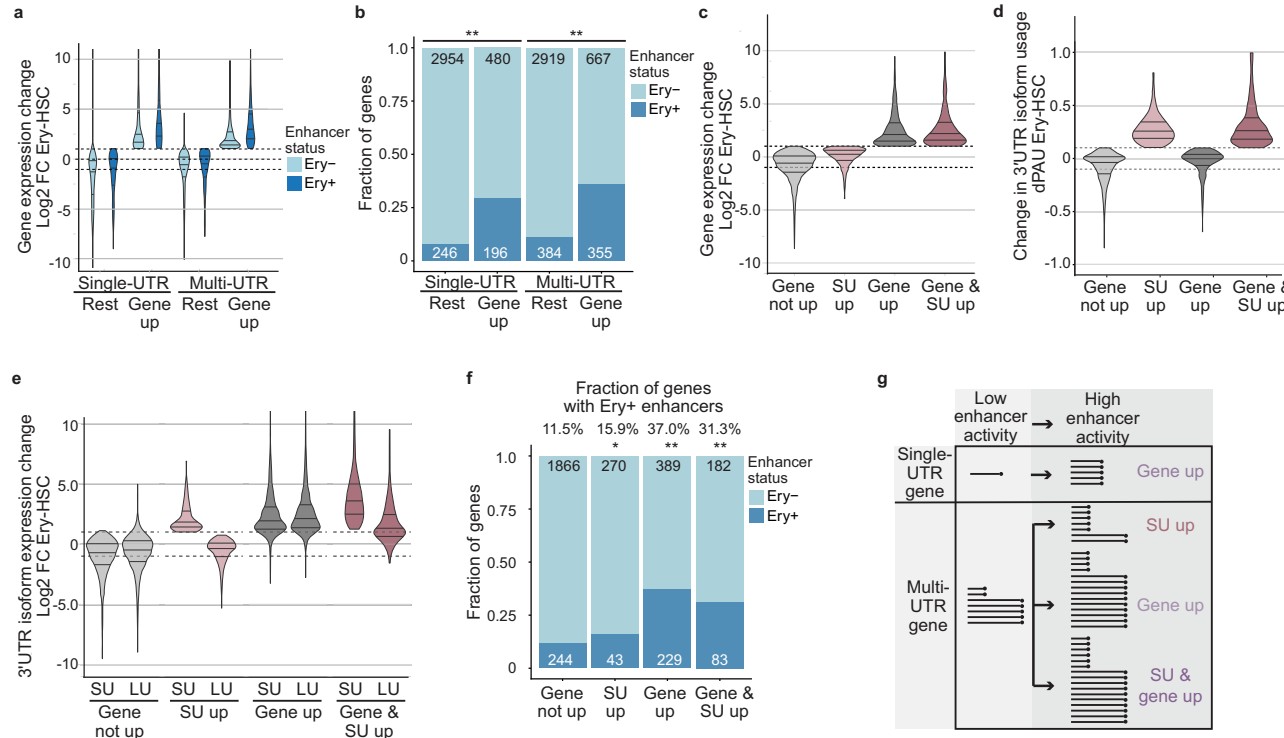

**Fig. 5 Cell type-specific enhancers are associated with genes that upregulate *SU* isoforms. a** Fold change (FC) in gene expression between erythroblasts (Ery) and hematopoietic stem cells (HSC) is shown for single- and multi-UTR genes separately. The genes that associate with erythroblast-specific enhancers (Ery+) are shown separately. Violin plots denote median, 25th and 75th percentiles. The number of genes in each group is shown in (**b**). **b** Fraction of genes associated with erythroblast-specific enhancers (Ery+) in the groups from (**a**). Chi-square tests were performed. Single-UTR genes: $X^2 = 249$, **$P < 10^{-16}$. Multi-UTR genes: $X^2 = 293$, **$P < 10^{-16}$. Ery-, genes without an erythroblast-specific enhancer in the vicinity. **c** Multi-UTR genes were separated based on their gene expression change using DESeq2 with a minimum FC > 2 into two groups ("gene up" and "gene not up"). These groups were further subdivided based on significant upregulation of their *SU* isoforms ("SU up" requires an absolute and relative increase (TPM FC > 2 and dPAU > 0.1, DEXseq, *P* < 0.05, 10% FDR), whereas the control group requires *SU* isoforms not to be upregulated (TPM FC < 2 and dPAU < 0.1, DEXseq, *P* > 0.05). FC in gene expression in the indicated groups is shown as in (**a**). The number of genes in each group is shown in (**f**). The PAU is a measure introduced by QAPA and indicates the relative usage of a 3′UTR isoform. Here, we only consider *SU* isoform usage for the PAU. dPAU is the differential PAU. **d** Change in 3′UTR isoform ratio as determined by a change in PAU for the groups from (**c**). **e** FC in 3′UTR isoform expression shown for *SU* and *LU* isoforms in the groups from (**c**). **f** Fraction of genes associated erythroblast-specific enhancers (Ery+) in the groups from (**c**). Chi-square tests were performed in comparison with the "gene not up" group: "SU up", $X^2 = 4.3$, *$P = 0.038$; "gene up", $X^2 = 215$, **$P < 10^{-16}$; "gene and SU up", $X^2 = 75.7$, **$P < 10^{-16}$. **g** Model showing enhancer-mediated increase in gene expression for single-UTR genes (top). Enhancer-mediated effect for multi-UTR genes can result in increased 3′UTR isoform expression, increased gene expression or an increase in both parameters. The lines with the black dots signify processed transcripts that contain a poly(A) tail.

promoters derived from multi-UTR genes (Fig. 3). As the enhancer-dependent increase in isoform expression was associated with decreased read-through transcription downstream of weak polyadenylation sites, we conclude that enhancers regulate 3′ end processing activity (Fig. 2). Our reporter data are further supported by results obtained at endogenous gene loci, where enhancer deletion of the multi-UTR gene *PTEN* did not change gene expression but altered 3′UTR isoform expression. Furthermore, KD of the transcription factor p65 which binds to the enhancer did not change steady-state mRNA level but induced a 3′UTR ratio change (Fig. 1). Moreover, in a transcriptome-wide analysis, we found that cell type-specific enhancers were significantly associated with genes that, during differentiation, exclusively upregulated their short 3′UTR isoforms without leading to a significant overall increase in gene expression (Fig. 5). However, as upregulation of *SU* isoforms usually results in a slight increase in gene expression, *SU* upregulation and a change in gene expression cannot be fully disentangled at endogenous genes (Supplementary Fig. 5a–d).

Our study further revealed that transcription and transcription elongation factors are responsible for increased transcript

processing at weak polyadenylation sites and subsequent upregulation of mRNA isoform expression (Fig. 4). Moreover, as suppression of histone acetyltransferase complexes decreases CPA activity, our data suggest that active enhancers regulate 3′ end processing (Fig. 4). However, the exact mechanism by which enhancers regulate CPA activity is currently unknown as we did not detect a linear relationship between overall gene expression and *SU* isoform usage across cell types (Supplementary Fig. 5e). Based on the literature, several potential mechanisms exist that are not mutually exclusive. It is established that RNA-binding proteins that bind to the polyadenylation signal and the surrounding sequence determine 3′ end CPA activity[7,8,16–22,59,61,73]. One model by which these RNA-binding proteins bind to polyadenylation sites is through the promoter loading model: Active enhancers recruit these factors to promoters which allows them to travel with RNA polymerase II and to bind to a newly transcribed polyadenylation site, thus increasing 3′ end processing activity locally[49–53,55–58]. Such a mechanism has been proposed for promoter-dependent regulation of post-transcriptional processes in yeast, including the regulation of mRNA stability, cytoplasmic localization, and translation[74–78]. This model is supported by the

presence of a variety of RNA-binding proteins at 80% of human promoters and by the observation that many RNA-binding proteins bind to transcription factors[55,56].

Another model suggests that active enhancers regulate transcription elongation rate which could result in differential usage of polyadenylation sites[23,24,79–81]. This is supported by our data showing that silencing of several factors involved in transcription elongation, including SPT4/5, NELF, and the PAF complex affect CPA activity of proximal polyadenylation sites (Table 1)[82]. Finally, the integrator complex has been shown to associate with active enhancers and increases enhancer-promoter communication[83]. Decreased expression of INTS11, the catalytic subunit of the complex, promotes read-through transcription at polyadenylation sites and shifts alternative isoform expression towards the distal isoform[32,84]. This suggests that increased association of integrator at active enhancers could prevent read-through and could increase CPA activity. Integrator is also known to regulate transcription elongation[85,86], but it is currently unclear if its role in transcription elongation is required for integrator-dependent regulation of 3′ end processing of protein-coding genes.

Currently, the in vivo usage of a given PAS cannot be predicted. Intrinsically weak sites, meaning PAS that are surrounded by a poor sequence context may get upregulated upon binding of suitable factors, whereas intrinsically strong PAS may have a lower potential for regulation. This is supported by our reporter results which revealed that enhancers or transcription factors upregulate CPA activity of 6/6 tested proximal PAS, whereas their influence on distal or single-UTR PAS was unpredictable (Fig. 2f). This work, together with work from others suggests that in vivo usage of PAS is regulated by the sequence context as well as by transcription factors, transcription elongation factors, chromatin regulators and expression levels of RNA-binding proteins, including CPA factors[7,8,23–25,40,82].

Although APA is widespread and is regulated in a cell type- and condition-specific manner, for most genes the consequences of a change in alternative 3′UTR isoform expression are unknown[3,7,8]. Nevertheless, striking examples exist in the literature that revealed that altering alternative 3′UTR isoform expression can result in substantial changes in protein expression caused by 3′UTR-dependent control of mRNA stability or translation[36,87,88]. However, alterations in alternative 3′UTR isoform expression often do not change overall protein levels (Fig. 1)[12,14,20,89,90]. For those cases, it has been demonstrated that isoform-specific differences in protein localization or function occur through 3′UTR-mediated formation of alternative protein complexes[11–15]. For example, the ubiquitin ligase BIRC3 switches from predominant *SU* isoform expression in normal B cells to *LU* isoform expression in malignant B cells. Normal B cells mostly form protein complexes that are independent of *LU* isoforms and they mediate BIRC3's tumor-suppressive functions. In contrast, malignant B cells preferentially form long 3′UTR-dependent protein complexes that have tumor-promoting roles[12]. So far, most studies that investigated the functional consequences of alternative 3′UTR isoform expression have relied on using expression constructs, but more recently CRISPR-mediated deletions of 3′UTRs were added to the tool kit[11–15,28,89–91]. Alternatively, as we showed here, 3′UTR isoform expression can be altered through enhancer deletion (Fig. 1). This strategy will allow researchers to study the resulting functional consequences at endogenous gene loci and has the advantage of keeping 3′UTR cis-elements intact while only changing the relative expression of 3′UTR isoforms. Although we only generated a heterozygous deletion of the *PTEN* enhancer, it was sufficient for detecting a molecular phenotype as shown by a switch in 3′UTR isoform expression. The switch induced by the heterozygous deletion was only partial. In an ideal case, we expect that enhancer activation

will result in SU-only expression, whereas full deletion of the enhancer will result in LU-only expression.

3′UTR length has expanded substantially during evolution of more complex animals and it correlates with the number of cell types observed in an organism[92,93]. At the same time, the number of enhancers has increased with organismal complexity[1]. Therefore, we speculate that increased regulation by enhancers has co-evolved with increased regulation by 3′UTRs to integrate intrinsic and extrinsic signals to change gene and mRNA isoform expression[2].

## Methods

**Cells used**. For all experiments, the human breast cancer cell line MCF7 was used which was a gift from the laboratory of Robert Weinberg (Whitehead Institute, Cambridge, USA). For the experiments with luciferase reporters integrated into the genome, the MCF7/FRT cell line was used which was a gift from the laboratory of Reuven Agami (Netherlands Cancer Institute, Amsterdam, Netherlands)[95]. All cells were cultured in DMEM supplemented with 10% FCS and 1% Pen-strep (normal condition). For experiments using acidified media, MCF7 cells were cultured for 24 h in DMEM supplemented with HCl (pH = 6.5). MCF7 cells were transfected with Lipofectamine 2000 (Invitrogen).

*PTEN enhancer deletion using CRISPR-Cas9*. All primers are listed in Supplementary Table 2. The guide RNAs targeting the 5′ and 3′ ends of the *PTEN* enhancer were designed using GuideScan and cloned into pX330 as previously described[96,97]. MCF7 cells were transfected with pmaxGFP and the two pX330 plasmids. After 4 days, GFP-positive cells were sorted into 96-well plates and grown into single cell colonies. Enhancer deletion was tested by PCR and sequencing.

*Screening of cells with PTEN enhancer deletion*. Genomic DNA was isolated from each clone and the deletion of the *PTEN* enhancer was assessed by PCR with primers (PcE-FS2 and PcE-RS2) flanking the enhancer region and Sanger sequencing. As MCF7 cells are not diploid at the *PTEN* locus, the presence of the alleles with *PTEN* enhancer sequences was further assessed by qPCR using genomic DNA isolated from WT and dE cells to amplify two regions outside of the *PTEN* enhancer (chr10:89,618,718-89,618,843 and chr10:89,620,263-89,620,387) and two regions inside *PTEN* enhancer (chr10:89,621,562-89,621,699 and chr10:89,621,320-89,621,440) using FastStart universal SYBR green master mix (Roche). The qPCR results from the amplicons inside the *PTEN* enhancer were normalized to those from the amplicons outside of the *PTEN* enhancer and were compared between WT and enhancer deletion cells to determine the extent of enhancer loss.

*ShRNA-mediated knock-down*. For the knock-down of *RELA* and *KLLN*, lentiviral vectors (pLKO-puro) containing shRNAs were purchased from Sigma (RELA, TRCN0000014684, TRCN0000014686; KLLN, TRCN0000339049, TRCN 0000339051). As control, an shRNA with scrambled sequence was used (Addgene, #1864)[98].

**Northern blotting**. Northern blotting was performed as described previously with modifications[87], dx.doi.org/10.17504/protocols.io.bqqymvxw. Briefly, total RNA was isolated using Tri reagent (Invitrogen). PolyA+ mRNA was purified with Oligotex (Qiagen) and 2 µg of polyA+ mRNA was loaded in each lane. The single-stranded probe was generated by unidirectional PCR reaction as described[99] with a slight modification. As input for the unidirectional PCR 30 ng of DNA template was used. This consisted of a 724-bp fragment of the *PTEN* coding region amplified from MCF7 cDNA (PTEN-NB-F and PTEN-NB-R). The unidirectional PCR reaction (20 µl) was conducted in buffer (100 mM Tris-HCl/pH 8.3, 500 mM KCl, 15 mM MgCl$_2$, 1% Triton-X-100) containing 0.2 mM each of dCTP, dGTP, and dTTP, 0.5 unit of Taq polymerase, the reverse primer from above, and 6 µl of 3000 Ci/mmol [$\alpha$-$^{32}$P]dATP (Perkin Elmer). The reaction mixture was initially boiled for 10 min at 95 °C and subjected to 35 thermal cycles (95 °C for 30 s; 45 °C for 30 s; 72 °C for 1 min), which was followed by 5 min incubation at 72 °C. After the PCR reaction, 5 µl of 0.2 mM EDTA was added to the mixture and boiled for 5 min at 95 °C, followed by 2 min chilling on ice. The PCR product was then used for hybridization to probe the *PTEN* transcript and the blot was scanned by Fuji phosphorimager.

**Quantification of endogenous *PTEN* transcripts**. For qRT-PCR, total RNA was isolated using Tri reagent (Invitrogen). cDNA synthesis was performed using qScript cDNA supermix (Quantabio), followed by qRT-PCR was performed using FastStart universal SYBR green master mix (Roche). *RPL19* mRNA was used as loading control.

To quantify steady-state total *PTEN* mRNA level, primers PTEN-qP-F and PTEN-qP-R were used. To quantify the fraction of *PTEN-LU* over total *PTEN* mRNA, PTENLU-qP-F and -R primers were used for RT-qPCR. The relative

abundance of *PTEN-LU* using this primer pair corresponds to the relative abundance of *PTEN-LU* obtained by northern blot analysis (e.g., Fig. 1f). To obtain the fraction of *PTEN-LU*, the abundance of *PTEN-LU* was normalized to the abundance of total *PTEN* mRNA detected by primers PTEN-qP-F and -R.

*Half-life of* PTEN-LU. MCF7 WT and dE2 cells were treated with or without 25 mM HCl overnight. The cells were then treated with either DMSO or 4 μg/ml actinomycin D for the indicated time points. The half-lives of *PTEN-LU* mRNA in each condition were determined as described previously[100,101]. Briefly, the amounts of *PTEN-LU* mRNA remaining were determined at each time point relative to that at 0 h. The relative amounts of *PTEN-LU* mRNA over time were subjected to nonlinear regression, given that the amount of *PTEN-LU* remaining at a certain time point can be calculated by the equation: remaining amount = $e^{kt}$ (where k and t are decay rate and time, respectively). The half-lives were then calculated by the equation: half-life = $\ln(0.5)/k$.

PTEN *splicing pattern*. A primer pair (PTEN-sp-F and PTEN-sp-R) located in the first and last exon of *PTEN* (NM_000314) was used on the cDNA described above and visualized on 1% agarose gels.

*Transcript production rate for endogenous* PTEN. The nascent transcripts were extracted following the protocol described by ref. [102] with minor modifications. Briefly, MCF7 cells were treated with 400 μM 4-thiouridine for 2 h (4sU, Fisher Scientific). After labeling, Tri reagent (Invitrogen) was used to extract total RNA from cells, followed by DNase I treatment (NEB). Next, 50 μg of total RNA was mixed with 0.5 μg of 4sU-labeled yeast RNA as a spike-in. The mixture was biotinylated with 10 μg of MTSEA biotin-XX (Biotium) in 100 mM HEPES/pH 7.5 and 10 mM EDTA for 30 min at 25 °C in dark with gentle agitation and the biotinylated RNA was recovered by ethanol precipitation. Then, half of the biotinylated RNA was mixed with 100 μM DTT and retained as the total RNA fraction. The other half was incubated with Dynabeads M-280 Streptavidin (Invitrogen) in bead wash buffer (100 mM Tris-HCl/pH 7.5, 1 M NaCl, 0.1% Tween-20, 10 mM EDTA), washed twice and eluted twice in 100 μl of 0.1 M DTT at 25 °C for 5 min, followed by ethanol precipitation. cDNA generated using Superscript III reverse transcriptase (Invitrogen) and random hexamers (Invitrogen). *PTEN* transcript abundance was measured by qPCR using an amplicon in the first intron of the *PTEN* gene (PTEN-int1-qP-F and PTEN-int1-qP-R) normalized to the abundance of yeast *ACT1* transcripts from the spike-in RNA (using primers yACT1-qP-F and yACT1-qP-R). The rate of transcript production was determined by the ratio of the abundance of the nascent *PTEN* transcripts to the total *PTEN* transcripts.

**Western blotting**. Cell pellets were lysed in 2x Laemmli buffer (Alfa Aesar) and performed as described previously[12] with the following antibodies: anti-PTEN (A2B1, Santa Cruz Biotechnology, sc-7974, 1:1000), anti-GAPDH (V-18, Santa Cruz Biotechnology, sc-20357, 1:500), anti-P65 (L8F6, Cell Signaling Technology, 6956, 1:1000), and anti-phospho-P65 Ser536 (93H1, Cell Signaling Technology, 3033, 1:1000). As secondary antibodies anti-mouse IRDye 800 (1:5000; Li-Cor Biosciences, Cat# 926-68072), anti-rabbit IRDye 680 (1:5000; Li-Cor Biosciences, Cat# 926–6807), and anti-goat IRDye 680 (1:5000; Li-Cor Biosciences, Cat# 926-32224) were used. The blot was analyzed by Odyssey CLx imaging system (Li-Cor).

**Luciferase reporter assays**
*Luciferase reporter constructs*. All luciferase constructs were derived from PIS1 vector[87]. It contains the thymidine kinase promoter of Herpes simplex virus, followed by a renilla luciferase open reading frame, followed by the late SV40 PAS. To obtain reporter constructs with different promoters or PAS, the promoter or the SV40 PAS were exchanged using restriction enzyme digest or Gibson cloning. To obtain reporter constructs with different enhancers, the enhancer sequences (Supplementary Table 1) were cloned upstream in the sense orientation of the respective promoters. In the case of Penh1, the reverse complement of the sequence was cloned downstream of the PAS. MYC-binding sites (E-boxes with the sequence CACGTG at positions −1149 bp and −1353 bp upstream of the transcription start site of the *PTEN* gene) were mutated to CAAGAA using Quikchange Lightening kit (Agilent). The TATA synthetic promoter sequence was derived from pGL firefly reporter with a minimal promoter.

*Luciferase assay*. Luciferase assays were performed in 24-well plates as described previously[87]. The number of experiments listed in the figures corresponds to biological replicates. In each well, 100 ng of firefly luciferase control reporter plasmid PIS0 together with 400 ng of renilla luciferase plasmid were transfected. Same molar amounts of plasmid were transfected to account for different construct sizes (400 ng were used for a plasmid of 5000 bp). Firefly and renilla luciferase activities were measured with the Dual-luciferase assay (Promega) 24 h after transfection using Glomax 96 microplate luminometer and Glomax 96 software (Promega). Renilla activity was normalized to firefly activity to control for transfection efficiency. Transcriptional activity of a promoter corresponds to renilla luciferase activity (normalized by firefly luciferase activity) after transfection of the reporter containing the promoter and the SV40 PAS. CPA activity of a test PAS

was obtained by dividing the luciferase activities of the constructs with the test PAS by the SV40 PAS in the context of the same promoter.

To assess CPA activity after knock-down of transcription factors, luciferase constructs were transfected into MCF7 cells stably expressing control (ctrl) shRNAs or shRNAs targeting specific transcription factors or co-activators. PAS usage was calculated as described above. When several shRNAs against a specific factor were available, the results were pooled. The fold change in transcriptional activity upon factor knock-down was obtained by normalization to the transcriptional activity of the SV40 PAS reporter in cells expressing ctrl shRNAs that were performed in parallel.

*mRNA stability of the reporter constructs*. Luciferase plasmids were transfected as described above. After 24 h, cells were either treated with DMSO or with actinomycin D (4 μg/ml; Sigma) for the indicated time points. Total RNA was extracted using Tri reagent and was used to generate cDNA using qScript qScript cDNA supermix (Quantabio). qRT-PCR was performed using FastStart universal SYBR green master mix (Roche) on a 7500 HT Fast Real-Time PCR System (Applied Biosystems). The primer pairs used to quantify total *PTEN* mRNA (PTEN-stability-F and -R), luciferase mRNA (Rluc-stability-F and -R) and *GAPDH* mRNA (GAPDH-F and GAPDH-stability-R) (for normalization) are listed in Supplementary Table 2.

*5′ RACE*. Renilla luciferase plasmids were transfected as described above and total RNA was extracted after 24 h using Tri reagent (Invitrogen). 5′ RACE was performed with the 5′ RACE kit (Roche) using gene-specific reverse primers: PTEN-5′RACE-R and R2 (for nested PCR).

*Measurement of read-through transcripts of the reporter constructs*. Luciferase reporter constructs were linearized by BglII, which cuts upstream of the enhancer/ promoter in the renilla luciferase constructs. 650 fmol of the linearized plasmids were transfected as described above. Total RNA was extracted after 24 h using Tri reagent (Invitrogen), followed by DNase I (NEB) treatment and ethanol precipitation. 1 μg of total RNA was used to generate cDNA using Superscript III reverse transcriptase (Invitrogen) and random hexamers (Invitrogen). Read-through transcripts were detected with a primer pair that localizes to the vector backbone (1040 bp downstream of the PAS; RenRT-qP-F and RenRT-qP-R). Read-through values were normalized to total transcript levels obtained by a primer pair that localizes to the renilla open reading frame (941 bp upstream of the PAS; RenORF-qP-F and RenORF-qP-R).

*Measurement of rolling-circle transcription in luciferase reporter constructs*. Luciferase plasmids were transfected as described above and total RNA was extracted after 24 h using Tri reagent, followed by DNase I treatment and ethanol precipitation. 1 μg of total RNA was used to generate cDNA using Superscript III reverse transcriptase and random hexamers. To detect the transcripts produced in vivo in a manner dependent on rolling-circle transcription on the renilla luciferase plasmids, PCR was conducted using a primer pair (Ren-Rolling-F and -R) that binds 147-bp upstream of BglII site which marks the start of the Penh. The total transcript levels were determined by PCR using a primer pair that localizes to the renilla open reading frame (Rluc-stability-F and RenORF-Rolling-R). The PCR products were resolved in 2% agarose gel and quantified using Multi-Gauge (Fuji). The amounts of the rolling-circle transcription were normalized to the total transcript levels.

*Small-scale shRNA screen*. shRNAs were designed using the siRNA selection program from the Whitehead Institute and cloned into pSUPERretropuro. Retroviral particles were obtained as described before[87]. Knock-down efficiency was tested by RT-PCR with gene-specific primers and primers for *GAPDH*.

*Luciferase assay using reporters integrated into the genome*. The expression cassettes of the renilla luciferase reporters (from Pprom or Penh-Pprom to PAS (SV40 PAS or PTEN PPAS) in PIS1 plasmids) were inserted by Gibson cloning into pcDNA5-FRT plasmid digested by BglII and SphI. The integration of each pcDNA5-FRT plasmid with the renilla luciferase reporters into MCF7-FRT cells was carried out by Flp-in system as described elsewhere[95]. Luciferase assays were conducted as described above after transfection of the cells with PIS0 plasmid.

**Association of SU usage and gene expression**. To assess if high levels of gene expression are associated with increased SU isoform usage, a dataset generated by Lianoglou et al. (2013) was reanalyzed[3]. The 10% highest and lowest expressed multi-UTR genes in each cell type were selected and the short 3′UTR index (SUI) which is the fraction of *SU* isoform expression divided by the total 3′UTR isoform expression, was plotted.

**Association of cell type-specific enhancers with alternative 3′UTR isoforms**
*Dataset selection*. To obtain genes regulated by cell type-specific enhancers, we used a publicly available dataset generated by Cai et al. (2020) who determined cell type-specific enhancers in murine definitive erythroblasts (here: Ery) and assigned them to target genes using a combination of methods including transcriptome analysis,

chromatin accessibility, histone modifications, transcription factor occupancy, and 3D chromatin interactions[68]. We used the assigned target genes from Supporting Information Dataset_S01.

To identify genes that increase their expression in erythroblasts, we compared gene expression between erythroblasts and murine hematopoietic stem cells (HSC). We set out to use replicates (two HSC datasets[69,94] and two Ery datasets[68,70]). The accession numbers are listed above. To perform quality control for the samples, we aligned them to mm10 using HISAT2 v2.1.0. Gene body coverage was estimated against the mm10 housekeeping gene annotation provided by RSeQC using the geneBody_coverage.py script from RSeQC v4.0.0[103]. MultiQC v1.10.1 was used to compare coverages[104]. This analysis revealed uneven coverage among the samples (Supplementary Fig. 5f). Uneven coverage will confound the analysis on 3′UTR isoform expression as QAPA determines 3′UTR isoform ratios by using the coverage of reads that fall into the region of the *SU* and *LU* isoforms. Moreover, the different datasets use different library preparation methods (pair-end reads, single-end reads, stranded, not stranded) which result in technical differences across the samples independent of their cell type-specific differences. To identify technically similar datasets, we performed principal component analysis (using the top 1000 genes of highest variance in the gene expression matrix after DESeq2's rlog transformation) on the four datasets. We observed that the datasets generated by Ling (2019) and Lee (2018) had a minimal distance in PC2 which reflects technical variation (Supplementary Fig. 5g)[69,70]. Therefore, these samples were used for downstream pairwise analysis as representative Ery and HSC cell types, respectively.

*Differential gene and 3′UTR isoform analysis.* To identify genes with a significant difference in gene expression between Ery and HSC, we used DESeq2 v1.28.0[105,106]. To identify genes with a significant difference in 3′UTR isoform expression, we first used QAPA to identify multi-UTR genes and to obtain TPM values for *SU* and *LU* isoforms[107]. Samples were pseudoaligned for transcript quantification in Salmon v1.3.0 ("salmon quant–gcBias–validateMappings -l A") to the pre-compiled QAPA v1.3.0 mm10 3′UTR annotation with the full mm10 genome as decoy[108]. Multi-UTR genes used in the analysis were filtered based on Num_Events > 1 (more than one annotated isoform) and at least 3 TPM in one or more samples. Single-UTR genes have Num_Events = 1. *SU* isoforms were identified by QAPA APA_IDs ending in "_P". To identify statistically significant changes in 3′UTR isoform usage we required a dPAU > 0.1 with a significant isoform ratio change according to DEXSeq v1.34.0 with 10% FDR. Normalized expression values were computed by rescaling QAPA TPM values per sample by estimated size factors from DESeq2. We further required a minimal fold change >2 for *SU* isoform expression to be considered significantly upregulated. All analysis was performed in R v4.0.2 and Bioconductor v1.13.

*Multi-UTR gene categories.* We first separated the multi-UTR genes into two categories based on their gene expression changes using DESeq2 ("gene up", "gene not up") using 10% FDR and at least 2-fold change in gene expression in erythroblasts. Next, we subdivided these groups with respect to "SU up" and identified the genes with at least 2-fold upregulation of *SU* isoform expression using QAPA TPM values and a significant 3′UTR isoform usage change with dPAU values obtained by QAPA of >0.1 and being significant according to DEXseq.

*Stratified random sampling.* The range of gene expression log2 fold-changes was subdivided into 13 strata of equal width, with 1 as an upper bound and the minimum of the "SU up" genes as the lower bound (approximately −4). For each stratum, we randomly subsampled the two groups ("gene not up" and "SU up") to have matching numbers of genes. The number of genes per stratum was determined by the group with the least genes in that stratum.

**Statistics**. For all pairwise comparisons of PAS usage or transcriptional activity a 2-sided, 2-sample unequal variance t-test for independent samples was applied. When comparing several samples, a One-way ANOVA was performed. Mann–Whitney tests were used to determine if gene expression influences SU usage. Chi-square ($X^2$) tests were used to test for significant enrichment of genes associated with Ery-specific enhancers. The Pearson value was reported. Statistical tests were performed using Excel, R, and SPSS.

**Reporting summary**. Further information on research design is available in the Nature Research Reporting Summary linked to this article.

## Data availability

The data supporting the findings of this study are available from the corresponding authors upon reasonable request. To identify the enhancers and promoters used in this study, levels of acetylated H3K27 and transcription factor binding sites in MCF7 cells were visualized using published ChIP-seq data (GSM946850), generated by the Encode project[43,44]. Binding of MYC to the *PTEN* promoter was assessed by using published ChIP-seq data (GSE33213). FASTQ files for bulk RNA-seq samples were obtained from the Sequence Read Archive. Mouse definitive erythroblasts: SRR6946157-9[68],

SRR8945139-41,44-45[70], mouse hematopoietic stem cells: SRR7946616-7[94], SRR6458998-9000[69]. 3′-seq data were obtained from SRP029953[3]. Source data for the figures and Supplementary Figures are provided as a Source Data file.

## Code availability

Our code is available at https://github.com/Mayrlab/utr-enhancers.

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

## Acknowledgements

We thank all members of the Mayr lab for helpful discussions and Kristian Helin for critical reading of the manuscript. This work was funded by the NIH Director's Pioneer Award (DP1-GM123454), Damon Runyon-Rachleff Innovation Award, the Pershing Square Sohn Cancer Research Alliance, and the NCI Cancer Center Support Grant (P30 CA008748).

## Author contributions

B.K. designed and performed all experiments regarding endogenous PTEN. B.K. measured read-through of the reporter and performed experiments on the integrated reporter, M.M.F. performed all computational analyses, N.P. and J.L. cloned the reporter constructs and performed the reporter assays, J.L. evaluated the knock-down efficiency of the shRNAs, W.M. performed the mRNA stability experiment of the reporter. C.M. conceived and supervised the project, designed and analyzed the reporter experiments, and wrote the manuscript with input from all authors.

## Competing interests

The authors declare no competing interests.
