## [Peer Review File · Nature Communications]

Title: Enhancers regulate 3' end processing activity to control expression of alternative 3'UTR isoformsREVIEWER COMMENTS

Reviewer #1 (Remarks to the Author):

In this manuscript from Kwon et al., the authors report a general impact of enhancer sequences on cleavage and polyadenylation (CPA) activity. This effect leads to alternative polyadenylation (APA) isoform changes in genes that have multiple polyA sites. Overall, the authors have done substantial amount experimental work and bioinformatic analysis. The data presented largely support their conclusions. There are, however, a number of issues the authors need to address before this manuscript can be accepted.

Major:

1. Much of the work was carried out by using plasmid-based reporter assays. Because plasmids are circular, RNA polymerase II can in theory go more than one round on the plasmid. Therefore, it would be difficult to rule out the possibility that the enhancing element functions merely to block RNA polymerase II, which could also alter reporter gene expression and APA site choice, from active participation in 3' end processing. Therefore, the authors should put the reporter constructs into the genome for assays. This should be done for at least 1-2 of their plasmids. Having the reporter in the genome could also increase the signal to noise ratio for readthrough analysis by RT-qPCR (Figs. 2d-2e), which can be substantially affected by transfected plasmid DNA (despite DNase treatment).
2. The impact of enhancer and promoter activities on 3' end processing and APA has been previously reported by several labs, including Rosonina...Blencowe, JBC, 2003; Nagaike...Manley, Mol Cell 2011; Ji...Tian, Mol Syst Biol, 2011. These works should be cited, so that readers are aware of the context of current investigation.

Minor:

1. The authors use 'transcriptional activity' and 'CPA activity' to indicate luciferase activities in different contexts. This could cause confusion. These terms are secondary based on luciferase activities, which are the primary readout. As such, it is recommended that the authors use 'luciferase activity' as y-axis labels in their plots. They could, however, indicate 'transcriptional activity' and 'CPA activity' in parenthesis as their interpretation of the luciferase activity.
2. Figure 1f. The authors claim that they did not detect an enhancer-or condition-specific difference in stability of the mRNA isoforms. However, it seems that there is some difference between WT and dE2 in normal conditions for the PTEN-LU (Fig. 1f, right). It would be helpful if they could generate half-life values based on decay curve and provide p-values for comparisons.

Reviewer #2 (Remarks to the Author):

This manuscript titled “Enhancers regulated 3’ end processing activity to control expression of alternative 3’UTR isoforms”, by Kwon et al, provides a novel perspective of transcriptional enhancers’ role in 3’end control. This is a new exciting idea and research at the intersection between the fields of transcription and RNA biology. The experiments and analyses appeared properly designed and conducted, although there may be a few alternative ways to explain their experimental observations. If the authors can effectively rule out these alternative hypotheses and consolidate their enhancer mediated 3’end regulation hypothesis, it would be a great fit for Nature Communications.

Major points.

1. Can the alterations in 3’ processing be a secondary effect of the changes in primary transcription? In other words, are SU/LU isoform ratios simply regulated by transcription level itself, and not necessarily by the enhancers? One analysis I suggest is to compare the expression levels of SU and LU isoforms in genes that are strongly expressed vs weakly expressed. If SU/LU expression ratios are positively correlated with the overall expression level transcriptome-wide, that may suggest that simply having higher transcription levels lead to higher SU expression. If not, this alternative hypothesis may be less likely and would better support the authors’ hypothesis.
2. On human genome annotation, PTEN is paired with the KLLN gene on the upstream antisense orientation. The region the authors deleted (Extended Data Fig 1) includes the coding sequence of the KLLN gene. KLLN protein is known to be expressed and may affect replication in MCF7 cells. Can the effects seen in Fig 1 be the secondary effect of KLLN gene knock-out or haploinsufficiency? This can be tested by using KLLN RNAi instead of CRISPR deletion. Absence of PTEN SU isoform increase under KLLN protein depletion using RNAi will confirm that deleting the enhancer elements on the KLLN gene is associated with PTEN isoform change.
3. In the reporter assay results, it will be necessary to rule out the interference from PTEN enhancer transcription or KLLN gene transcription. It will be particularly important if the authors used a method using transient transfection of circular plasmids. A run-through transcription from the Penh may affect the transcription initiation and transcript processing on the plasmid template. This concern is somewhat consistent with their finding that Penh1 was not increasing the transcription from the Pprom, but Penh1rc positioned at the downstream of the Rluc was increasing the Rluc expression (Extended Data Fig 2e,g). If the eRNA run-through interferes with the Rluc expression upstream of the promoter, the reversed downstream transcription orientation may have relieved it. It is also possible that there is rolling circle transcription along the plasmid. It needs to be validated that eRNA run-through is not present or significant in their reporter system. A simple reverse transcription PCR using site specific primers at the intervening regions of the circular plasmids may be able to confirm this. If there is significant read-through transcription, it may be necessary to insert strong PASs flanking the Penh1 in the reporter system.
4. The siRNA screening results of TFs and Co-activators RNAi screening is overall in the direction of supporting their hypothesis, but the details are confusing. Many factors show increased transcription after the siRNA treatment in Fig 4, which is contrary to the expectation. I suspect this would be related

to how the data is normalized. The promoter of the control Firefly luciferase (SV40 promoter?) may also have been affected by the TF and Co-A KDs, and may have not served as a proper normalization control. It might be worth normalizing for transfected plasmid copy numbers, or if the transfection efficiencies were relatively uniform, they could present the non-normalized data. This point may not affect the author's conclusions but will better explain their siRNA experiments.

5. In Fig 5, the authors show that cell type specific enhancers are associated with altered 3' end processing in an erythroid differentiation model. The difference in the Ery+ enhancer fractions between the Rest vs SU up genes (Fig 5e) is the critical comparison supporting their claim. It may be possible that these SU up genes are in fact transcriptionally increased, but not large enough to be classified as Gene up. In Fig5c, the SU up group shows slightly higher gene expression than the Rest group. It will be necessary to more rigorously control the gene expression level in the Rest class vs SU up class to be more comparable. For example, the authors can use a more rigorous cut-off to select the SU group, or use statistical techniques such as stratified random sampling (SRS) to match the expression level of the Rest class to the SU up class. They should still be able to find statistically significant enrichment of Ery+ enhancers in the SU up genes compared to the SRS controlled Rest genes.

6. Also, it would be more convincing if the authors could perform a similar cell type specific enhancer analysis in another cell type-specific RNA-seq dataset from publicly accessible in datasets, such as GTEX.

Minor point

In Fig 1c,1d, since these are negative results, proper activation of NF-kb and MYC by the acid treatment needs to be validated, for example, by testing p65 phosphorylation.

Reviewer #3 (Remarks to the Author):

In the study by Kwon et al, the authors investigate the role that enhancers play in polyA site selection. Using the PTEN enhancer as an initial model, they utilize CRISPR to delete the enhancer allowing for the detection of a isoform switch upon acidification. This observation is characterized using luciferase reporters and is extended to other PASs of different strengths and to different enhancers. Several transcription factors and co-activators are identified as potentially playing a role in this regulation and the authors finish using functional genomics to determine how widespread this phenomenon may be throughout the genome. The strength of this paper lies within its novelty as it has not been seen that enhancers have this property while the weakness lies in several aspects of the experiments and interpretation. To recommend publication, the authors are encouraged to address through writing and experimentation my following concerns:

1. Figure 1: The authors use CRISPR to create two clonal MCF7 cell lines where one copy of the PTEN

locus lacks the PTEN-enhancer and the other one retains the copy of the enhancer. By their own description, these two lines are heterozygous and the Figure 1b clearly confirms this (assuming that MCF7 cells are diploid for this gene). Where I am deeply confused is how to interpret the results as the authors describe this section as if they have created a homozygous PTEN-enhancer deletion line. Evidence for this is in the following – “fully abrogated in cells lacking the enhancer”, “PTEN enhancer is required for a..”. Shouldn’t these sentences really read ‘fully abrogated in cells lacking ONE enhancer’ or “BOTH copies of the PTEN enhancer are required for a...”? Can the authors clarify this issue in the text and discuss why a homozygous line was not attained or isn’t necessary?

2. Figure 1: while it is compelling that acidified media creates an isoform switch, it is not as clear that this is mediated through myc or NFkB. Can the authors also conduct a depletion experiment of either of these factors +/- acid or at least show that acid increases their occupancy of the PTEN enhancer through CHIP? The concern here is the potential for pleiotropy accompanied by broad changes in pH.

3. Along the same lines as my point #2, have the authors tested the effect of acidified media on their luciferase reporters? This would also confirm that effects seen in Figure 1 are indeed mediated through the PTEN enhancer.

4. Figure 2e: I am a little confused what samples were compared for the statistical significance as the bars above are a little off. Shouldn’t the authors be making two sets of comparisons: RT transcript levels SV40PAS +/-Penh and also RT transcript levels for PTEN PPAS +/- the Penh (like in panel 2C)? The way the authors describe this in results, they claim to ‘observe 3-fold more RT in this reporter....’ This sentence is vague and requires that the authors specifically state which reporter is being compared to which (as not all comparisons are significant). If the authors are meaning to compare the PTEN PPAS +/- the Penh, then the data might not be significant – then what does this mean? Authors need to be clearer here.

5. Figure 2f: The constructs presented here do not appear to be systematic. The authors test the NUDT21 PPAS and the DPAS but only test the DICER PPAS and not the Dicer DPAS – why? Rather, they present the PTEN DPAS. Additionally, the results are not all discussed or interpreted – the NUDT21 DPAS CPA activity decreases due to the enhance as does GAPDH whereas others are not impacted. This needs to be clarified in some manner. Also, the authors need to provide a schematic with sequences of the multiple PASs to display what some are considered ‘strong’ versus ‘weak’. A non-expert would have no way to grasp this point.

6. Figure 2f: have the authors considered the less confounding alternative to using different strengths of PASs from different genes and simply mutating the PTEN PPAS to make is stronger/more consensus?

7. Figure 3: The narrative of this study is that the PTEN enhancer stimulates the CPA efficiency of the PTEN PPAS making it unexpected that the authors did not test whether the NUDT21 putative enhance stimulates the CPA efficiency of the NUDT21 PPAS. Based upon the constructs shown in Figure 2f, this

seems like the most logical experiment to conduct.

8. Figure 4: It may be helpful to clearly define (re-define) how transcriptional activity and CPA are measured here. As it stands, it is confusing why depletion of a large number of transcription factors or co-activators appears to increase the transcriptional activity relative to control – this seems counter-intuitive. It would help if the authors used two distinct positive controls in this assay as the FIP1L KD made it super easy to understand what a leftward shift relative to control would mean. But, there is not positive control for transcription factor KD to understand which direction on the y-axis things should be shifting. Also, can the authors comment as to whether this mini shRNA screen was done in the presence of acidified media? Would seem that it should have in order to sensitive cells PTEN PPAS regulation.

9. Fig 4C: the authors point out MYC and RELA as relevant hits in the shRNA screen and reference figure 4C – but I do not see them highlighted? Maybe they are referring to 4D?

10. If CPA is defined as the amount of luciferase produced from a test PAS relative to SV40 PAS (gold standard), then I would recommend the authors consider reconstructing their figure schematics. In many cases (ie Figures 2a, 3b, and 4b) the schematics have SC40 on top as opposed to the tested PAS – this could be misleading to the reader.

REVIEWER COMMENTS

Reviewer #1 (Remarks to the Author):

In this manuscript from Kwon et al., the authors report a general impact of enhancer sequences on cleavage and polyadenylation (CPA) activity. This effect leads to alternative polyadenylation (APA) isoform changes in genes that have multiple polyA sites. Overall, the authors have done substantial amount experimental work and bioinformatic analysis. The data presented largely support their conclusions. There are, however, a number of issues the authors need to address before this manuscript can be accepted.

Major:

1. Much of the work was carried out by using plasmid-based reporter assays. Because plasmids are circular, RNA polymerase II can in theory goes more than one round on the plasmid. Therefore, it would be difficult to rule out the possibility that the enhancing element functions merely to block RNA polymerase II, which could also alter reporter gene expression and APA site choice, from active participation in 3' end processing. Therefore, the authors should put the reporter constructs into the genome for assays. This should be done for at least 1-2 of their plasmids. Having the reporter in the genome could also increase the signal to noise ratio for readthrough analysis by RT-qPCR (Figs. 2d-2e), which can be substantially affected by transfected plasmid DNA (despite DNase treatment).

We thank the reviewer for spending time to review our manuscript and for the insightful comments that strengthen our paper. As suggested, we integrated the most important reporters into the genome of MCF7 cells using the Flp-in system. We integrated the Pprom and the Penh-Pprom reporters terminated by the SV40 PAS or the PTEN PPAS. As shown in the new Supplementary Fig. 2c and 2d, the *PTEN* enhancer increased the relative CPA activity of the PTEN PPAS also when integrated into the genome.

We tried to measure read-through transcription on the integrated reporter. However, in the Flp-in cell line, downstream of the PAS of the insert, an FRT site, the lacZ gene, and the zeocin resistance gene are located that were used to generate the Flp-in locus. These elements have a GC content of 55-70% and we were unable to find a primer pair in the downstream region that generates a specific PCR product. Moreover, the higher GC content may act as pause site for RNA polymerase II (Gromak et al., 2006, PMID: 1488997), thus increasing CPA activity of weak PAS. This may be the reason for the slightly higher CPA activity of PTEN PPAS in the context of the Pprom when it was integrated into the genome.

To obtain more reliable results when measuring read-through of the transfected reporters, we included more biological replicates (see updated Fig. 2e).

2. The impact of enhancer and promoter activities on 3' end processing and APA has been previously reported by several labs, including Rosonina...Blencowe, JBC, 2003; Nagaike...Manley, Mol Cell 2011; Ji...Tian, Mol Syst Biol, 2011. These works should be cited, so that readers are aware of the context of current investigation.

We had already cited the Rosonina and Nagaike papers and added the Ji paper in the revised manuscript.

Minor:

1. The authors use 'transcriptional activity' and 'CPA activity' to indicate luciferase activities in different contexts. This could cause confusion. These terms are secondary based on luciferase activities, which are the primary readout. As such, it is recommended that the authors use 'luciferase activity' as y-axis

labels in their plots. They could, however, indicate ‘transcriptional activity’ and ‘CPA activity’ in parenthesis as their interpretation of the luciferase activity.

We like this suggestion very much and have changed the y-axes according to the reviewer’s suggestion.

2. Figure 1f. The authors claim that they did not detect an enhancer-or condition-specific difference in stability of the mRNA isoforms. However, it seems that there is some difference between WT and dE2 in normal conditions for the PTEN-LU (Fig. 1f, right). It would be helpful if they could they generate half-life values based on decay curve and provide p-values for comparisons.

As suggested, we performed additional biological replicates and determined the half-lives of the PTEN-LU transcripts in the four conditions. We plotted the mRNA isoform decay curves from three biological replicates in the new Fig. 1i. This analysis shows that the decay kinetics of the PTEN-LU transcript are not significantly different in normal vs acidic conditions or in the presence or heterozygous absence of the *PTEN* enhancer.

Reviewer #2 (Remarks to the Author):

This manuscript titled “Enhancers regulated 3’ end processing activity to control expression of alternative 3’UTR isoforms”, by Kwon et al, provides a novel perspective of transcriptional enhancers’ role in 3’end control. This is a new exciting idea and research at the intersection between the fields of transcription and RNA biology. The experiments and analyses appeared properly designed and conducted, although there may be a few alternative ways to explain their experimental observations. If the authors can effectively rule out these alternative hypotheses and consolidate their enhancer mediated 3’end regulation hypothesis, it would be a great fit for Nature Communications.

We want to thank the reviewer for spending time to review our manuscript and for the insightful remarks that improved our paper.

Major points.

1. Can the alterations in 3’ processing be a secondary effect of the changes in primary transcription? In other words, are SU/LU isoform ratios simply regulated by transcription level itself, and not necessarily by the enhancers? One analysis I suggest is to compare the expression levels of SU and LU isoforms in genes that are strongly expressed vs weakly expressed. If SU/LU expression ratios are positively correlated with the overall expression level transcriptome-wide, that may suggest that simply having higher transcription levels lead to higher SU expression. If not, this alternative hypothesis may be less likely and would better support the authors’ hypothesis.

To address this comment, we re-analyzed our previously published 3’-seq dataset (Lianoglou et al., Genes Dev 2013). For each sample, we selected the 10% highest and 10% lowest expressed multi-UTR genes and plotted the fraction of *SU* isoform expression. We added this analysis in Supplementary Fig. 5e. The analysis shows that the distribution of *SU* isoform ratio is very wide, but it does not significantly differ between the highest and lowest expressed genes.

2. On human genome annotation, PTEN is paired with the KLLN gene on the upstream antisense orientation. The region the authors deleted (Extended Data Fig 1) includes the coding sequence of the KLLN gene. KLLN protein is known to be expressed and may affect replication in MCF7 cells. Can the effects seen in Fig 1 be the secondary effect of KLLN gene knock-out or haploinsufficiency? This can be tested by using KLLN RNAi instead of CRISPR deletion. Absence of PTEN *SU* isoform increase

under KLLN protein depletion using RNAi will confirm that deleting the enhancer elements on the KLLN gene is associated with PTEN isoform change.

We agree with the reviewer that the presence of the *KLLN* gene is an important confounding variable. As suggested, we knocked-down the *KLLN* gene using two different shRNA and achieved a decent suppression at the mRNA level (new Supplementary Fig. 1f). We then measured expression of PTEN-LU/total PTEN mRNA in control knock-down and *KLLN* knock-down conditions and did not observe a difference (Supplementary Fig. 1g). This suggests that the deletion of the *KLLN* gene is not the cause for the observed change in PTEN 3'UTR isoform expression upon heterozygous *PTEN* enhancer deletion.

3. In the reporter assay results, it will be necessary to rule out the interference from PTEN enhancer transcription or *KLLN* gene transcription. It will be particularly important if the authors used a method using transient transfection of circular plasmids. A run-through transcription from the Penh may affect the transcription initiation and transcript processing on the plasmid template. This concern is somewhat consistent with their finding that Penh1 was not increasing the transcription from the Pprom, but Penh1rc positioned at the downstream of the Rluc was increasing the Rluc expression (Extended Data Fig 2e,g). If the eRNA run-through interferes with the Rluc expression upstream of the promoter, the reversed downstream transcription orientation may have relieved it. It is also possible that there is rolling circle transcription along the plasmid. It needs to be validated that eRNA run-through is not present or significant in their reporter system. A simple reverse transcription PCR using site specific primers at the intervening regions of the circular plasmids may be able to confirm this. If there is significant read-through transcription, it may be necessary to insert strong PASs flanking the Penh1 in the reporter system.

We agree with the reviewer that ruling out rolling circle transcription is important for proper interpretation of the reporter results. We determined rolling circle transcription between plasmids that contain the Pprom and the Penh-Pprom. In the presence of the Penh, we observed slightly lower (not significant) transcription at the intervening region of the plasmid (see new Supplementary Fig. 2g). This result excludes the possibility that transcripts from the Penh affect transcription initiation or processing.

Moreover, based on the suggestion of Rev#1, we integrated the reporters into the genome using Flp-in cells. As shown in the new Supplementary Fig. 2c and 2d, the *PTEN* enhancer increased the relative CPA activity of the PTEN PPAS also when integrated into the genome, without significantly changing transcriptional activity.

4. The siRNA screening results of TFs and Co-activators RNAi screening is overall in the direction of supporting their hypothesis, but the details are confusing. Many factors show increased transcription after the siRNA treatment in Fig 4, which is contrary to the expectation. I suspect this would be related to how the data is normalized. The promoter of the control Firefly luciferase (SV40 promoter?) may also have been affected by the TF and Co-A KDs, and may have not served as a proper normalization control. It might be worth normalizing for transfected plasmid copy numbers, or if the transfection efficiencies were relatively uniform, they could present the non-normalized data. This point may not affect the author's conclusions but will better explain their siRNA experiments.

We agree with the reviewer. Based on the suggestion of the reviewer, we revisited the way the reporter assay was normalized. We did not observe a consistent bias in expression of the control Firefly luciferase reporter upon TF knock-down. Therefore, we kept this normalization control to account for differences in cell numbers and transfection efficiency.

The screen was performed in several batches. Although transcriptional activity of the TF knock-down samples was consistent among biological replicates, the transcriptional activity of the ctrl knock-down samples differed among the batches. In the initial submission, transcriptional activity of ctrl knock-down samples obtained from early batches was used for normalization of the entire screen as the transcriptional activity was similar to the activity obtained in untransfected MCF7 cells. However, the appropriate control for transcriptional activity of the TF knock-down samples is the transcriptional activity of the ctrl knock-down samples from the same batch. In the revised version of the manuscript, we report the fold change in transcriptional activity observed in TF knock-down samples normalized to ctrl knock-down samples from the same batch. As expected by the reviewer, knock-down of TFs results in lower transcriptional activity. We revised Fig. 4c, Table 1, and Supplementary Fig. 4b accordingly.

5. In Fig 5, the authors show that cell type specific enhancers are associated with altered 3'end processing in an erythroid differentiation model. The difference in the Ery+ enhancer fractions between the Rest vs SU up genes (Fig 5e) is the critical comparison supporting their claim. It may be possible that these SU up genes are in fact transcriptionally increased, but not large enough to be classified as Gene up. In Fig5c, the SU up group shows slightly higher gene expression than the Rest group. It will be necessary to more rigorously control the gene expression level in the Rest class vs SU up class to be more comparable. For example, the authors can use a more rigorous cut-off to select the SU group, or use statistical techniques such as stratified random sampling (SRS) to match the expression level of the Rest class to the SU up class. They should still be able to find statistically significant enrichment of Ery+ enhancers in the SU up genes compared to the SRS controlled rest genes.

We agree with the reviewer that we need to more rigorously control for gene expression in the 'rest' group vs the SU up group. When revisiting the data, we noticed that the 'rest' group still contained genes whose *SU* isoforms are upregulated by more than 2-fold (they cross the dashed line in the original Fig. 5d). These genes should be in the SU up group. They were initially missed as their gene expression decreased. Therefore, we redefined our groups using more stringent criteria. A gene is considered as SU up, if the *SU* isoform is upregulated in absolute and relative terms, meaning that it needs to be upregulated by at least 2-fold with respect to its expression (TPM), but it also requires a significant 3'UTR ratio change towards upregulation of the *SU* isoform ($dPAU > 0.1$, DEXseq significant with 10%FDR) to make sure that the *LU* isoform is not simultaneously upregulated. The control group should not have SU up, meaning that both criteria (TPM fold change of $SU < 2$ and $dPAU < 0.1$) should be met. Genes that only meet one of the criteria were excluded. This analysis reduces the number of genes in the analysis, thus reducing our statistical power, but our conclusions remain the same. The data is shown in Fig. 5c-f.

In our new analysis, the 'rest' (which was renamed 'gene not up') group still has a lower gene expression pattern than the SU up group (Fig. 5c). Therefore, as suggested by the reviewer, we used stratified random sampling to match the gene expression level of the 'SU up' group. To fully exclude any gene with a slight upregulation of *SU*, we applied stringent criteria. We required the control genes to upregulate their *SU* isoforms by less than 1.25-fold in TPM and required the 3'UTR ratio change towards *SU* upregulation to be less than 5% ($dPAU < 0.05$; Supplementary Fig. 5a-d). The stringent criteria make sure that genes in the control group do not upregulate their *SU* isoforms, but this reduces the number of genes available for comparison. For three stratified random samples, our analysis shows that the fraction of genes associated with Ery+ enhancers is lower in the control group (mean, 12.2%) than in the SU up group (mean, 15.4%). However, as the Chi-square test is very sensitive to the number of genes tested, the small sample number does not reach statistical significance (Supplementary Fig. 5d).

This analysis reveals that it is difficult to disentangle an upregulation of gene expression from exclusive upregulation of *SU* isoforms as this usually also slightly upregulates gene expression. We mention this limitation of our analysis in the results section and in the Discussion.

6. Also, it would be more convincing if the authors could perform a similar cell type specific enhancer analysis in another cell type-specific RNA-seq dataset from publicly accessible in datasets, such as GTEX.

We agree with the reviewer. We searched for additional datasets, but to be useful, they need to have annotated cell type-specific enhancers and the RNA-seq datasets for the cell types to be compared need to have a similar quality and need to be performed with the same library preparation protocols. As we use QAPA to estimate 3'UTR isoform usage, the RNA-seq read distribution needs to be comparable as otherwise the obtained results are skewed. Sadly, none of the candidate datasets passed all our quality controls.

Minor point

In Fig 1c,1d, since these are negative results, proper activation of NF-kb and MYC by the acid treatment needs to be validated, for example, by testing p65 phosphorylation.

As suggested, we tested if media acidification induces NF-κB activation by blotting for phosphorylated p65. We show in the new Fig. 1e (and in the new Supplementary Fig. 1c) that media acidification induces p65 phosphorylation, suggesting that media acidification activates the *PTEN* enhancer. Moreover, knock-down of p65 prevented the *PTEN* 3'UTR ratio change of endogenous *PTEN* without affecting steady-state *PTEN* mRNA level (new Fig. 1g, 1h). These experiments strongly suggest that the p65 transcription factor regulates *PTEN* 3'UTR isoform expression of the endogenous gene. This finding supports our overall finding that enhancers together with the bound transcription factors regulate 3' end processing activity.

Reviewer #3 (Remarks to the Author):

In the study by Kwon et al, the authors investigate the role that enhancers play in polyA site selection. Using the *PTEN* enhancer as an initial model, they utilize CRISPR to delete the enhancer allowing for the detection of a isoform switch upon acidification. This observation is characterized using luciferase reporters and is extended to other PASs of different strengths and to different enhancers. Several transcription factors and co-activators are identified as potentially playing a role in this regulation and the authors finish using functional genomics to determine how widespread this phenomenon may be throughout the genome. The strength of this paper lies within its novelty as it has not been seen that enhancers have this property while the weakness lies in several aspects of the experiments and interpretation. To recommend publication, the authors are encouraged to address through writing and experimentation my following concerns:

We want to thank the reviewer for spending time to review our manuscript and for the insightful comments that made our paper stronger.

1. Figure 1: The authors use CRISPR to create two clonal MCF7 cell lines where one copy of the *PTEN* locus lacks the *PTEN*-enhancer and the other one retains the copy of the enhancer. By their own description, these two lines are heterozygous and the Figure 1b clearly confirms this (assuming that MCF7 cells are diploid for this gene). Where I am deeply confused is how to interpret the results as the authors describe this section as if they have created a homozygous *PTEN*-enhancer deletion line. Evidence for this is in the following – “fully abrogated in cells lacking the enhancer”, “*PTEN*

enhancer is required for a..”. Shouldn’t these sentences really read ‘fully abrogated in cells lacking ONE enhancer’ or “BOTH copies of the PTEN enhancer are required for a...”? Can the authors clarify this issue in the text and discuss why a homozygous line was not attained or isn’t necessary?

We agree with the reviewer that the text describing these clones is not correct. In the revised manuscript we describe them more carefully and mention that the obtained clones are heterozygous.

As we already observed a phenotype from the heterozygous deletion of the *PTEN* enhancer with respect to a change in 3'UTR isoform expression of *PTEN*, we refrained from performing a second round of single clone isolation to generate homozygous deletions. In our experience, the generation of single cell clones can have downsides by inducing collateral changes in gene expression. A second round of single cell clone generation may potentiate the problems. Therefore, we prefer to work with heterozygous deletion clones.

2. Figure 1: while it is compelling that acidified media creates an isoform switch, it is not as clear that this is mediated through *myc* or *NFKB*. Can the authors also conduct a depletion experiment of either of these factors +/- acid or at least show that acid increases their occupancy of the *PTEN* enhancer through CHIP? The concern here is the potential for pleiotropy accompanied by broad changes in pH.

As suggested, we tested if media acidification induces activation of *NF-κB* (as was reported by others). We show in the new Fig. 1e (and in the new Supplementary Fig. 1c) that media acidification induces p65 phosphorylation, indicative of *NF-κB* activation. Next, we used two different shRNAs to knockdown p65 as it was shown by ChIP-seq to bind to the *PTEN* enhancer. Knock-down of p65 did not affect steady-state *PTEN* mRNA level (new Fig. 1g). However, suppression of p65 prevented the acidification-induced *PTEN* 3'UTR ratio change (new Fig. 1h). These experiments strongly suggest that the p65 transcription factor regulates *PTEN* 3'UTR isoform expression of the endogenous gene. This finding supports our overall finding that enhancers together with the bound transcription factors regulate 3' end processing activity.

3. Along the same lines as my point #2, have the authors tested the effect of acidified media on their luciferase reporters? This would also confirm that effects seen in Figure 1 are indeed mediated through the *PTEN* enhancer.

Because of the potential pleiotropic effects of pH changes, we refrained from using acidified media in the reporter assays. Without acidification, we already observed strong differences in CPA activity in the presence or absence of different enhancers. The reporter assay differs in several aspects from the endogenous context. 1) The *PTEN* enhancer is either fully present or absent (and not present in a heterozygous manner), 2) transfection achieves much higher levels of the reporter gene than what is observed in the endogenous situation. These reasons may contribute to the differences seen in the reporter compared with the endogenous gene.

4. Figure 2e: I am a little confused what samples were compared for the statistical significance as the bars above are a little off. Shouldn’t the authors be making two sets of comparisons: RT transcript levels SV40PAS +/-Penh and also RT transcript levels for *PTEN* PPAS +/- the Penh (like in panel 2C)? The way the authors describe this in results, they claim to ‘observe 3-fold more RT in this reporter....’ This sentence is vague and requires that the authors specifically state which reporter is being compared to which (as not all comparisons are significant). If the authors are meaning to compare the *PTEN* PPAS +/- the Penh, then the data might not be significant – then what does this mean? Authors need to be clearer here.

We apologize for the confusing representation of our data. We compare read-through transcripts of a pair of reporters that are transcribed from the same promoter. We observe a similar amount of read-through transcripts downstream of the SV40 PAS or the *PTEN* PPAS when the reporters were

transcribed from the Penh-Pprom constructs. However, we observe 3-fold higher amount of read-through transcripts downstream of the PTEN PPAS compared to the SV40 PAS when the reporters were transcribed from the Pprom promoter. This means that in the absence of the *PTEN* enhancer, we observe more read-through downstream of the PTEN PPAS which is consistent with lower luciferase expression from this construct. We performed additional biological replicates to make our results more robust and revised the text describing Fig. 2e to clarify our observations.

5. Figure 2f: The constructs presented here do not appear to be systematic. The authors test the NUDT21 PPAS and the DPAS but only test the DICER PPAS and not the Dicer DPAS – why? Rather, they present the PTEN DPAS. Additionally, the results are not all discussed or interpreted – the NUDT21 DPAS CPA activity decreases due to the enhance as does GAPDH whereas others are not impacted. This needs to be clarified in some manner. Also, the authors need to provide a schematic with sequences of the multiple PASs to display what some are considered ‘strong’ versus ‘weak’. A non-expert would have no way to grasp this point.

After having observed increased CPA of the proximal PAS of PTEN, we selected two additional proximal PAS, two distal PAS and two PAS from single-UTR genes. The sequences are given in Supplementary Table 1a. At the time of experimentation, it was not possible to infer the strength of a PAS from its sequence. In the meantime, a paper describing a neural network was published whose scores are in concordance with our observations (Bogard et al., Cell 2019, PMC6599575).

To choose a PAS for testing in our assay, we selected proximal PAS that are ‘weak’ as they contribute to less than 30% of *SU* isoform expression in most cell types analyzed, according to our 3’UTR isoform expression study across 14 cell types and tissues (Lianoglou et al., Genes Dev 2013). However, the PAS strength of distal or single-UTR genes cannot be inferred from isoform expression levels. For example, a distal PAS could be strong, but if a preceding proximal PAS is also strong, its usage may appear low. Also, expression of single-UTR genes is determined by transcription and PAS strength, but the two parameters cannot be disentangled at endogenous genes. Nevertheless, we expected that distal PAS and single-UTR PAS would be strong and may not be regulatable by enhancers.

However, the reporter assay revealed that nearly all PAS are susceptible to a change in usage. As the direction of enhancer-mediated change for distal or single-UTR PAS was unpredictable, we did not include more such sites in subsequent assays. Instead, we focused on proximal PAS as they are considered weaker and as all showed an enhancer-dependent increase in CPA activity. We added a few sentences to the results and discussion section where we mention the results of the distal and single-UTR PAS and where we address the current limitations of estimating PAS strength.

6. Figure 2f: have the authors considered the less confounding alternative to using different strengths of PASs from different genes and simply mutating the PTEN PPAS to make is stronger/more consensus?

Most functional PAS contain the hexamer AAUAAA. This sequence is required for the CPA machinery to recognize that this is a potential mRNA 3’ end. As this element is essential, it cannot be mutated. The sequence surrounding the hexamer is considered the sequence that regulates the strength. However, the rules that determine PAS strength are currently unknown. Therefore, it is currently impossible to predict the outcome of mutations in the surrounding sequence.

7. Figure 3: The narrative of this study is that the PTEN enhancer stimulates the CPA efficiency of the PTEN PPAS making it unexpected that the authors did not test whether the NUDT21 putative enhance stimulates the CPA efficiency of the NUDT21 PPAS. Based upon the constructs shown in Figure 2f, this seems like the most logical experiment to conduct.

We wanted to test if a distal enhancer would also be able to change PAS usage. Although the Denh is in the vicinity of the *NUDT21* gene, we do not have evidence that it is actually an enhancer of the *NUDT21* gene. Nevertheless, as suggested by the reviewer, we performed a reporter assay where we tested the influence of the Denh on the *NUDT21* PPAS. In the new Supplementary Fig. 2l and 2m, we show that the enhancer increases PAS usage by 1.8-fold.

8. Figure 4: It may be helpful to clearly define (re-define) how transcriptional activity and CPA are measured here. As it stands, it is confusing why depletion of a large number of transcription factors or co-activators appears to increase the transcriptional activity relative to control – this seems counter-intuitive. It would help if the authors used two distinct positive controls in this assay as the FIP1L KD made it super easy to understand what a leftward shift relative to control would mean. But, there is not positive control for transcription factor KD to understand which direction on the y-axis things should be shifting. Also, can the authors comment as to whether this mini shRNA screen was done in the presence of acidified media? Would seem that it should have in order to sensitive cells PTEN PPAS regulation.

We agree with the reviewer and a similar point was raised by Rev#2. Based on the suggestions of the reviewer, we revisited the way the reporter assay was normalized. We double-checked if transcription factor knock-down would influence the expression of the transfection control (Firefly) luciferase reporter, but we did not observe a systematic bias.

The screen was performed in several batches. Although transcriptional activity of the TF knock-down samples was consistent among biological replicates, the transcriptional activity of the ctrl knock-down samples differed among the batches. In the initial submission, transcriptional activity of ctrl knock-down samples obtained from early batches was used for normalization of the entire screen as the transcriptional activity was similar to the activity obtained in untransfected MCF7 cells. However, the appropriate control for transcriptional activity of the TF knock-down samples is the transcriptional activity of the ctrl knock-down samples from the same batch. In the revised version of the manuscript, we report the fold change in transcriptional activity observed in TF knock-down samples normalized to ctrl knock-down samples from the same batch. As expected, knock-down of TFs results in lower transcriptional activity. We revised Fig. 4c, Table 1, and Supplementary Fig. 4b accordingly.

As stated above, all the luciferase assays were performed in normal media.

9. Fig 4C: the authors point out MYC and RELA as relevant hits in the shRNA screen and reference figure 4C – but I do not see them highlighted? Maybe they are referring to 4D?

We thank the reviewer for pointing out this discrepancy. In the new Fig. 4c, we are highlighting a few factors from each class of regulators, including MYC and RELA.

10. If CPA is defined as the amount of luciferase produced from a test PAS relative to SV40 PAS (gold standard), then I would recommend the authors consider reconstructing their figure schematics. In many cases (ie Figures 2a, 3b, and 4b) the schematics have SC40 on top as opposed to the tested PAS – this could be misleading to the reader.

We revised the figure schematics as suggested.

REVIEWERS' COMMENTS

Reviewer #1 (Remarks to the Author):

The authors have addressed all my concerns. I have no more comments/questions.

Reviewer #2 (Remarks to the Author):

In the revised manuscript by Kwon et al, most of the concerns I raised in the initial review were addressed well. The authors effectively ruled out many of the alternative possibilities. There is 1 major point that remains, which could be resolved if there are additional clarifications made.

1. Concern on potential association with transcription level and CPA efficiency is sufficiently addressed with Fig S5e

2. Concern on potential secondary effects of KLLN is sufficiently addressed in Fig S1f,g.

3. Concern on upstream antisense or rolling circle transcription of eRNA transcription is partially resolved by Fig S2g, although a few questions still remain.

1) How is rolling circle transcription quantified and normalized? From the methods, the RT-PCR assay seems to be not strand specific. My concern was the upstream antisense transcription that can directly collide with reporter transcription more often at the downstream regions. Could the rolling circle quantification detect both sense and antisense transcription at the indicated primer site?

2) This concern could also have been addressed if the authors transfected with linearized plasmid reporters. On a second reading of the manuscript and the methods section, I found that read-through transcript quantification was done using BglII linearized plasmid, but it is unclear how the luciferase assays were performed. Would it still be possible to add a linearized plasmid reporter experiment?

3) Examination under chromatinized context per reviewer 1's point 1 is obviously a more definitive experiment. Direct measurement of read-through transcripts in FLP-In integrated reporter would have resolved the concern to a large degree, but it is unfortunate that there were technical difficulties in the experiment. While the CPA reporter assay was statistically significant, there was also a slight (but NS) increase in Tx reporter assay. Would the analysis of the ratio of CPA/Tx still show significant differences? It might be helpful to add more repeats (currently n=3) to acquire stronger statistical power.

4 . Concern on the normalization of siRNA experiment details is now resolved after the authors explained how they revisited the normalization by the control assays.

5. Addition of the stratified random sampling analysis in Fig S5a-e relieves the concern. Although it was difficult to reach statistical significance, replication of the same trend is reassuring.

6. It is unfortunate that the authors could not find a suitable dataset to conduct a similar analysis as in Fig 5, which would have been a valuable addition.

Reviewer #3 (Remarks to the Author):

The authors have done an adequate job addressing my concerns. I still stand by the comments that heterozygous deletion of the enhancer as being sufficient to alter APA could use further clarification in the Discussion. Specifically, how the authors envision that deletion of one enhancer elicits what appears to be a 'dominant' effect on both alleles. This could very well be the case, but some additional language might be beneficial.

REVIEWERS' COMMENTS

Reviewer #1 (Remarks to the Author):

The authors have addressed all my concerns. I have no more comments/questions.

Reviewer #2 (Remarks to the Author):

In the revised manuscript by Kwon et al, most of the concerns I raised in the initial review were addressed well. The authors effectively ruled out many of the alternative possibilities. There is 1 major point that remains, which could be resolved if there are additional clarifications made.

1. Concern on potential association with transcription level and CPA efficiency is sufficiently addressed with Fig S5e

2. Concern on potential secondary effects of KLLN is sufficiently addressed in Fig S1f,g.

3. Concern on upstream antisense or rolling circle transcription of eRNA transcription is partially resolved by Fig S2g, although a few questions still remain.

1) How is rolling circle transcription quantified and normalized? From the methods, the RT-PCR assay seems to be not strand specific. My concern was the upstream antisense transcription that can directly collide with reporter transcription more often at the downstream regions. Could the rolling circle quantification detect both sense and antisense transcription at the indicated primer site?

As described in the Methods, the rolling-circle transcription was quantified by the primer pair that binds 147-bp upstream of Penh, and normalized to the total transcript level. As the reviewer pointed out, this RT-PCR is not strand-specific, and thus, if antisense transcription occurs, it will also be detected. In the case of collision between sense and potential anti-sense transcription, we expect degradation of both elongating transcripts, thus decreasing the amount of RT-PCR product. The difference between presence or absence of the Penh was not statistically significant (Fig. S2g); this result makes any potential effect of antisense transcription on CPA activity of the reporters unlikely. Moreover, as the reviewer pointed out in the question #3-2 below, we measured the levels of read-through transcription using linearized constructs and obtain consistent results (Fig. 2d-e). In the context of the linearized constructs, any effect of antisense transcription can be ruled out, indicating a regulatory role of the *PTEN* enhancer for CPA activity of the *PTEN* PPAS.

2) This concern could also have been addressed if the authors transfected with linearized plasmid reporters. On a second reading of the manuscript and the methods section, I found that read-through transcript quantification was done using BglII linearized plasmid, but it is unclear how the luciferase assays were performed. Would it still be possible to add a linearized plasmid reporter experiment?

We always used the circular-form plasmids for the luciferase assays as described in the Methods. However, following the reviewer #1's suggestion in the last revision, we conducted luciferase assays using reporters integrated into the genome. In this condition, we again observed the increase of CPA

activity of PTEN PPAS by Penh (Fig. S2c), which is in line with the findings based on the circularized plasmids.

3) Examination under chromatinized context per reviewer 1's point 1 is obviously a more definitive experiment. Direct measurement of read-through transcripts in FLP-In integrated reporter would have resolved the concern to a large degree, but it is unfortunate that there were technical difficulties in the experiment. While the CPA reporter assay was statistically significant, there was also a slight (but NS) increase in Tx reporter assay. Would the analysis of the ratio of CPA/Tx still show significant differences? It might be helpful to add more repeats (currently n=3) to acquire stronger statistical power.

We already took the transcription levels into account when comparing CPA activities in all the luciferase assays (plasmid-borne and integrated reporters). As described in the Methods, CPA activity of PTEN PPAS was obtained by dividing the luciferase activity of the construct with PTEN PPAS by that with SV40 PAS in the context of the same promoter. As a result, the CPA activities presented in the manuscript (for example, Fig. S2c for the integrated reporters) are already normalized to the transcription levels. Since a statistical significance was observed there, we did not include any more replicates.

4. Concern on the normalization of siRNA experiment details is now resolved after the authors explained how they revisited the normalization by the control assays.

5. Addition of the stratified random sampling analysis in Fig S5a-e relieves the concern. Although it was difficult to reach statistical significance, replication of the same trend is reassuring.

6. It is unfortunate that the authors could not find a suitable dataset to conduct a similar analysis as in Fig 5, which would have been a valuable addition.

Reviewer #3 (Remarks to the Author):

The authors have done an adequate job addressing my concerns. I still stand by the comments that heterozygous deletion of the enhancer as being sufficient to alter APA could use further clarification in the Discussion. Specifically, how the authors envision that deletion of one enhancer elicits what appears to be a 'dominant' effect on both alleles. This could very well be the case, but some additional language might be beneficial.

In our view, the effect of heterozygous enhancer deletion is only partial. In the presence of an active enhancer (presence of the sequence and activation by signaling), the ratio of PTEN-SU:LU is 90:10 (Fig. 1f; in an ideal case, the ratio would be 100:0, indicating expression of the SU isoform only). Full deletion of the enhancer should result in the opposite, namely PTEN-SU:LU of 0:100. Heterozygous deletion of the enhancer results in a partial effect, with a PTEN-SU:LU ratio of ~40:60, as shown in Fig. 1f. We added a few sentences to the discussion to point this out (line 413).